# Comprehensive re-analysis of hairpin small RNAs in fungi reveals loci with conserved links

Nathan R Johnson[1,2], Luis F Larrondo[1,3], José M Álvarez[1,2,4]*, Elena A Vidal[1,2,5]*

[1]Millennium Science Initiative - Millennium Institute for Integrative Biology (iBio), Santiago, Chile; [2]Centro de Genómica y Bioinformática, Facultad de Ciencias, Ingeniería y Tecnología, Universidad Mayor, Santiago, Chile; [3]Departamento de Genética Molecular y Microbiología, Facultad de Ciencias Biológicas, Pontificia Universidad Católica de Chile, Santiago, Chile; [4]Centro de Biotecnología Vegetal, Facultad de Ciencias, Universidad Andrés Bello, Santiago, Chile; [5]Escuela de Biotecnología, Facultad de Ciencias, Ingeniería y Tecnología, Universidad Mayor, Santiago, Chile

**\*For correspondence:**
jose.alvarez.h@unab.cl (JMÁ);
elena.vidal@umayor.cl (EAV)

**Abstract** RNA interference is an ancient mechanism with many regulatory roles in eukaryotic genomes, with small RNAs acting as their functional element. While there is a wide array of classes of small-RNA-producing loci, those resulting from stem-loop structures (hairpins) have received profuse attention. Such is the case of microRNAs (miRNAs), which have distinct roles in plants and animals. Fungi also produce small RNAs, and several publications have identified miRNAs and miRNA-like (mi/milRNA) hairpin RNAs in diverse fungal species using deep sequencing technologies. Despite this relevant source of information, relatively little is known about mi/milRNA features in fungi, mostly due to a lack of established criteria for their annotation. To systematically assess mi/milRNA characteristics and annotation confidence, we searched for publications describing mi/milRNA loci and re-assessed the annotations for 41 fungal species. We extracted and normalized the annotation data for 1727 reported mi/milRNA loci and determined their abundance profiles, concluding that less than half of the reported loci passed basic standards used for hairpin RNA discovery. We found that fungal mi/milRNA are generally more similar in size to animal miRNAs and were frequently associated with protein-coding genes. The compiled genomic analyses identified 25 mi/milRNA loci conserved in multiple species. Our pipeline allowed us to build a general hierarchy of locus quality, identifying more than 150 loci with high-quality annotations. We provide a centralized annotation of identified mi/milRNA hairpin RNAs in fungi which will serve as a resource for future research and advance in understanding the characteristics and functions of mi/milRNAs in fungal organisms.

## Editor's evaluation

This article is of interest to scientists within the field of RNA silencing and evolution. The data analysis is rigorous, and the conclusions are justified by the data. The key claims of the manuscript provide a compelling approach to identifying and annotating microRNAs in fungi.

## Introduction

Silencing by RNA interference (RNAi) is an ancient system for regulating RNA abundance found within eukaryotes. Small regulatory RNAs (sRNAs) are the functional elements behind RNAi and are typically

20–24 nucleotides in length. Functionally, sRNAs play key roles in genome stability (*Lewsey et al., 2016*; *Nolan et al., 2005*), protection against RNA-based organisms (*Nicolás and Ruiz-Vázquez, 2013*; *Segers et al., 2007*), and regulation of gene expression (*Jones-Rhoades et al., 2006*). In the context of pathogenic organisms, such as fungal plant pathogens, sRNAs have even been shown to display *trans*-kingdom functions (*Wang et al., 2017a*; *Weiberg et al., 2013*), where fungal sRNAs serve as effectors of pathogenicity regulating host genes to undermine resistance.

The biogenesis pathway of an sRNA locus can provide key insights into its function in an organism. Significantly, pathways differ by the sRNA class and by organism but generally follow the same schema (*Borges and Martienssen, 2015*; *Torres-Martínez and Ruiz-Vázquez, 2017*) (summarized in *Figure 1A*). Regions of double-stranded RNA (dsRNA) are processed by a Type-III RNAse called Dicer or Dicer-like (collectively abbreviated here as DCR), producing sRNA duplexes which are subsequently loaded in an Argonaute protein (AGO) for their resulting function. DCRs are responsible for the resulting sRNA length (*Macrae et al., 2006*) and these lengths are selective for the specific AGO loading. As a result, DCRs directly influence sRNA function (*Fang and Qi, 2016*). The source of the dsRNA is also critical to sRNA function. Two main pathways exist for this: (1) dsRNA produced through synthesis by an RNA-dependent RNA polymerase (RDR) and (2) complementary regions in transcripts form stem-loop foldback structures known as hairpins (*Axtell, 2013a*). Several RDR-derived sRNA classes have been defined in eukaryotes, with fungal types including those resulting from DNA damage (qiRNAs) (*Lee et al., 2009*), associated with meiotic silencing (*Hammond et al., 2013*), or associated with transcriptional silencing (i.e. quelling) (*Fulci and Macino, 2007*). Hairpin-derived sRNAs (hpRNAs) are widespread in eukaryotes and include microRNAs (miRNAs), which have central roles in gene expression control in plants and animals.

Highly precise dicing characterizes miRNAs, with the most-abundant sequences (MASs) coming from a single duplex in the 5' and 3' hairpin arms (*Kozomara and Griffiths-Jones, 2014*). These have been sometimes referred to as the mature-miRNA or the miR and the miR*, though these terms are imprecise considering both of these sequences may be functional and may reverse in the rank of abundance (*Li et al., 2012*; *Liu et al., 2017*; *Yang et al., 2011*). Homologs of miRNAs are classified into families, defined by nearly identical mature-miRNA sequences, though more distant relationships have been shown to exist (*Xia et al., 2013*). The mechanistic function of miRNAs is distinctly different by clade. In plants, miRNAs function mostly through direct cleavage of a target mRNA (*Rogers and Chen, 2013*), whereas in animals the process is less straightforward, relying on inhibition of translation and mRNA de-tailing, de-capping, and degradation by exonucleases (*Jonas and Izaurralde, 2015*). A significant number of miRNA are anciently conserved in sequence and function (*Cuperus et al., 2011*), though they may also be highly clade- or species-specific (*Fahlgren et al., 2010*; *Johnson et al., 2019*). Most miRNAs come from intergenic or untranslated genic regions (introns, UTRs), and there are rare if any examples of miRNAs derived from coding sequences (CDS) (*Liu et al., 2018*; *Olena and Patton, 2010*). There are also examples of hpRNAs that are clearly divergent from miRNAs and produce a spectrum of sizes from differing pathways (*Axtell, 2013a*). Long hairpin RNAs have long been a biotechnological tool for induction of RNAi (*Fusaro et al., 2006*), however, exploration is needed to identify clearly if this occurs in organisms naturally.

Fungi have also been shown to produce hpRNAs. The hpRNA class of miRNA-like RNAs (milRNAs) was first identified in *Neurospora crassa* (*Necra*), where they follow the same schema: RNA foldbacks are processed by one of two redundant DCRs (NcDCL1 and NcDCL2), which are subsequently loaded into an AGO protein (QDE2) (*Lee et al., 2010*). Homolog proteins of the DCR and AGO families have been identified in many fungal species (*Choi et al., 2014*) and are involved in milRNA biogenesis and function. Since, over 60 publications have cited hpRNAs in fungi, using the designations miRNA or milRNA (collectively referred to as mi/milRNAs). In fungi, these sRNAs have been identified frequently in the context of pathogenesis (*Xu et al., 2020*). This also includes some proposed to regulate gene expression in trans (*Cui et al., 2019*; *Wang et al., 2017b*; *Wong-Bajracharya et al., 2022*), regulating plant and animal host gene expression. However, most lack clear evidence for the causative role of the sRNA in the process, as many studies rely on target prediction alone for understanding their biological role (*Pinzón et al., 2017*).

The advent of small-RNA-seq technology has massively increased the amount of data available for detecting sRNA-producing loci. This has affected the standards of miRNA identification, as we now can rely on high-quality evidence for their identification, as opposed to low-throughput and

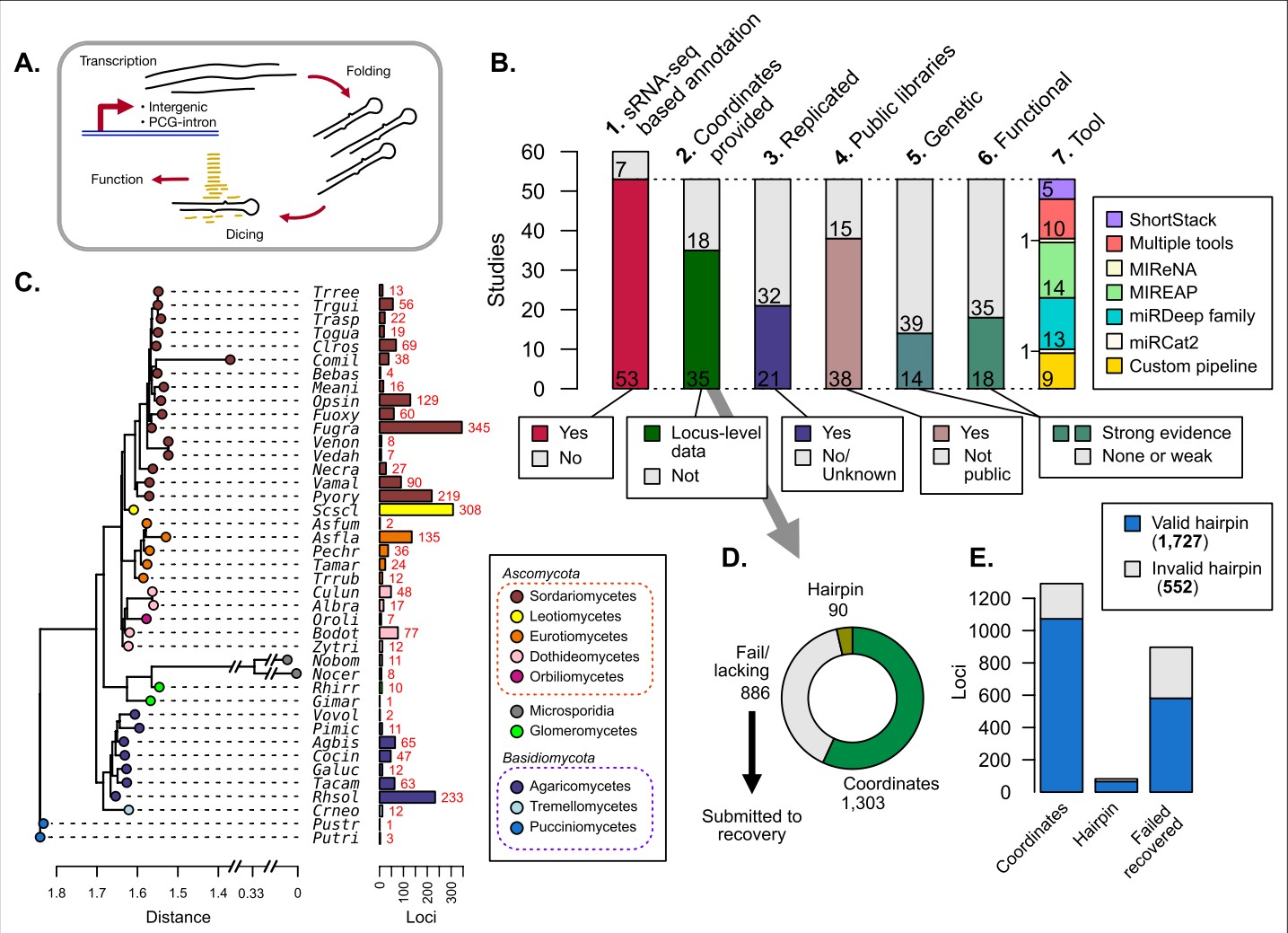

**Figure 1.** Publications referring to microRNAs (miRNAs) and/or miRNA-like (mi/milRNA) RNAs in fungi were identified, classified, and assessed for reported loci. (**A**) Generalized depiction of canonical miRNA synthesis pathway. (**B**) Breakdown of important metrics in fungal mi/milRNA loci identified in publications (***Supplementary file 1***). 1. Number of published studies reporting sRNA-seq-based annotations. 2. Number of publications where genomic coordinates are given for each mi/milRNA precursor gene. 3. Number of studies that have two or more library replicates for a condition. 4. Number of studies that make the sRNA-seq data available through a public repository. 5. Number of studies that support the synthesis pathway for the mi/milRNA through knock-out or over-expression lines. 6. Number of studies that give evidence for the function of an mi/milRNA through target-site manipulation or molecular evidence of cleavage (evidence quality designations described in Materials and methods). 7. Number of studies using a given tool/pipeline to discover mi/milRNA loci. (**C**) Dendrogram constructed from 18S rRNA sequences for the species with reported loci in A1 (MUSCLE alignment, RAxML maximum likelihood tree). Colored tips represent the taxonomic class for each species identified by SILVA. Names are given as a five-letter abbreviation (full names in ***Supplementary file 2***). Number of loci identified is shown by colored bars. (**D**) Assessment of the quality of loci reporting for published mi/milRNA loci. Loci are classified by whether they have a reported sequence and genomic coordinates for the entire precursor hairpin (coordinates, green), only reported sequence for the precursor hairpin without genomic coordinates (hairpin, olive green), or have insufficient or no information regarding the hairpin locus (fail/lacking, gray). Loci in this final category were subjected to a recovery pipeline to characterize their genomic coordinates, as explained in Materials and methods. (**E**) An assessment of hairpins relative to their source evidence as explained in C. Valid hairpins are defined as sequences containing the proposed most-abundant sequence, with no secondary structures or more than 20 unpaired bases in the duplex pairing region.

The online version of this article includes the following figure supplement(s) for figure 1:

**Figure supplement 1.** Analysis of locus count by citation.

**Figure supplement 2.** Accuracy for recovery pipeline.

**Figure supplement 3.** Flow diagram of confirmation pipeline for publications, miRNAs and miRNA-like (mi/milRNAs), sRNA-seq confirmation, and tier classification (produced using diagrams.net).

error-prone approaches like identification through RT-PCR. A wide spectrum of tools are available for annotation of miRNAs, with most of them based on strict rules derived from characteristics that are specific to known miRNAs from plants or animals (*Axtell and Meyers, 2018*; *Friedländer et al., 2012*; *Johnson et al., 2016*; *Kuang et al., 2019*). Despite many publications identifying mi/milRNA loci in fungi, there has yet to be a systematic assessment of their characteristics and annotation confidence, as in other eukaryotic species (*Kozomara and Griffiths-Jones, 2014*; *Meng et al., 2012*). In plants and animals, much is known about miRNAs. Here, we know the lengths of the hairpins and their foldback dynamics, the sizes of sRNAs that are produced, and the mechanisms behind their targeting (*Kozomara and Griffiths-Jones, 2014*; *Kozomara and Griffiths-Jones, 2011*). We also know the genomic context for the hairpins and the extent of their evolutionary conservation. In fungi, we lack this categorical knowledge about mi/milRNAs, which is further complicated by the abundance of non-uniform pipelines and inconsistent methodology, raising significant questions about the quality of their systematic identification. In this work, we focused solely on sRNAs derived from hairpin structures, aiming to identify fundamental characteristics of fungal mi/milRNAs. Furthermore, we sought to assess the quality of their annotations based on key metrics, ultimately building a centralized and well-documented annotation for public use. In this effort, we provide the first steps to evaluate the actual shape, size, context, and function of mi/milRNAs in fungi.

## Results

### Annotation of mi/milRNA in fungal species is sparse and heterogeneous

To gain insights into fungal mi/milRNA distinctive features and to assess the state of the genomic annotation of published loci, we conducted an exhaustive search for publications including terms related to fungal miRNAs and milRNAs in PubMed. The search yielded a list of 60 publications which assess fungal mi/milRNAs, 53 of which performed and reported de novo mi/milRNA loci identification from sRNA-seq data (*Figure 1B*, *Supplementary file 1*), using a broad range of bioinformatic tools or pipelines (*Figure 1B*). Noteworthy, these tools have been developed for miRNA prediction and annotation in plants and/or animals and can fail to identify loci with particular characteristics of mi/milRNA in fungi, as we demonstrated below. The majority of these sRNA-seq libraries are publicly available (*Figure 1B*, *Supplementary file 1*) and span 41 fungal species, including some subspecies (*Figure 1C*), presenting a prime opportunity to uncover common fungal mi/milRNA characteristics. These species come from a variety of lifestyles and relevance to humans, including laboratory models as well as many plant and animal pathogens (*Supplementary file 2*). Nearly all explored species are Ascomycetes and Basidiomycetes, though more distant divisions of Glomeromycota and Microsporidia are also included (*Figure 1C*, *Supplementary file 2*). Several species have many observed mi/milRNA loci, such as *Sclerotinia sclerotiorum* (*Scscl*) or *Fusarium graminearum* (*Fugra*) (*Figure 1C*). However, these counts may be more related to the study or the particular number of studies for a given organism (*Figure 1—figure supplement 1*). This could be due to differing levels of bioinformatic and/or experimental evidence required for mi/milRNA annotation and may fail to reveal their actual abundance/diversity in particular fungal species. For instance, only around half of the publications provide coordinates or complete sequences for precursor genes (*Figure 1B*), key information to derive structural and compositional features of the hairpins and genomic contexts. Furthermore, less than half of the published mi/milRNA have support from at least two independent biological replicates (*Figure 1B*), a requirement considered as vital for miRNA annotation (*Axtell and Meyers, 2018*). Regarding experimental validation, a minority of the source publications for a locus provide genetic evidence to support mi/milRNA biogenesis machinery using knock-out or over-expressor lines, or explore targeting relationships with direct experimental evidence (*Figure 1B*). This assessment indicates a need for establishing a set of standard rules for reporting mi/milRNA annotations in fungi, as it has been described for animals and plants (*Axtell and Meyers, 2018*; *Kozomara and Griffiths-Jones, 2014*).

### Loci lacking coordinates were recovered using genome-based inference

Reporting sequence information and genomic coordinates of mi/milRNA loci is essential to determine precursor features. Around one-third of loci had neither an annotation associated with the full precursor sequence nor complete genomic coordinates (*Figure 1D*). To gain insights into the genomic

origin of these hairpins, we utilized bowtie alignment to identify perfect matches of a supplied mature RNA sequence to the corresponding genome version and modeled possible precursor hairpin structures with RNAfold (*Lorenz et al., 2011*). This hairpin precursor recovery pipeline was evaluated in terms of precision and sensitivity for identifying the correct locality for an mi/milRNA locus as well as its folding pattern, confirming that our pipeline has reasonably high accuracy for predicting the correct genomic locality, based on the number of alignments for a cited MAS for an mi/milRNA (*Figure 1—figure supplement 2A*). We also found solid metrics for recapitulating the same hairpin when testing the pipeline against known hairpins (*Figure 1—figure supplement 2B*). We considered three minimal criteria for determining whether a sequence corresponds to a valid hairpin: (1) the precursor contains the reported MAS(s) within its boundaries, (2) the foldback region that gives rise to the RNA duplex must not contain secondary stems, and (3) the duplex must not contain large loops (maximum of 20 unpaired bases) (*Axtell and Meyers, 2018*). Hairpins reported in publications and obtained from the recovery pipeline were evaluated according to these standards, finding that most of them met these minimum requirements (*Figure 1E*). Notably, around 20% of the published loci failed one or more of these tests and were excluded from further analyses (further referred to as 'invalid').

## Independent re-assessment of reported mi/milRNA reveals inconsistent expression of expected most-abundant RNA sequences for an important proportion of loci

Current miRNA annotation pipelines depend on algorithms that mine high-throughput sRNA-seq data to determine mature miRNAs, as well as putative precursor sequences. These algorithms utilize different approaches to fulfill this task. The most commonly used tools perform or require a read alignment to genome sequences, followed by locus determination based on the genomic position and depth of the reads. Although bioinformatic prediction of sRNA loci from sequencing data is the preferred method to discover miRNAs and other types of sRNAs, frequently bona fide miRNA annotations require further curation and validation considering the differences in annotation standards. Hence, the quality of the annotations may appear quite variable among the submitted datasets (*Figure 1—figure supplement 1*; *Axtell and Meyers, 2018*; *Kozomara et al., 2019*; *Ludwig et al., 2017*; *Meng et al., 2012*).

We sought to independently evaluate reported mi/milRNAs in each publication, taking advantage of the availability of most of the source sRNA-seq data used for annotation (*Figure 1B*). Libraries were obtained from public repositories and trimmed, filtered, and aligned. We confirmed alignment rates mostly similar to that of their parent publications for each step, where the data are available (*Supplementary file 3*). In these publications, mi/milRNA are reported with a short sequence referred to as the mature sequence or alternatively as the miRNA or milRNA. In the case of animal and plant miRNA discovery, mature miRNA sequences are the MASs aligned to a locus, as a requirement of precise locus processing (*Axtell and Meyers, 2018*; *Kozomara and Griffiths-Jones, 2014*). However, within a large proportion of loci, we found that the mature sequence reported is not the MAS for the locus (*Figure 2A*, *Supplementary file 4*). Even more problematic, we found numerous examples of loci that had no reads aligning with the reported mature sequence or loci with no reads aligned at all (*Figure 2A*, *Supplementary file 4*). These categories are examples for which the locus annotation pipeline may need further curation via a consistent set of rules. Close inspection of sequences in raw, untrimmed libraries shows that it is unlikely these issues are caused by differences in the trimming, filtering, or alignment steps, since the exact reported mature mi/milRNA sequence was not present in the raw libraries. In loci where another aligned sequence was found to be the MAS, this sequence was used as the corresponding mature sequence in subsequent analyses (*Supplementary file 5*).

## Variable rates of confirmed mi/milRNA loci among published datasets

As mentioned earlier, several sets of rules have been established for miRNA discovery and annotation in plants and animals (*Friedländer et al., 2012*; *Johnson et al., 2016*; *Kozomara and Griffiths-Jones, 2014*; *Kuang et al., 2019*). These rules are derived from examples of known miRNA loci in well-studied organisms, and focus on the specific alignment profile of sRNA-seq reads over these loci, which is intimately related to the biogenesis mechanism of these regulatory molecules. Applying these criteria (summarized in *Supplementary file 6*), we found that fungal sRNAs frequently fail to meet key points compared to animal or plant miRNAs, resulting in very few that pass all rules in a set.

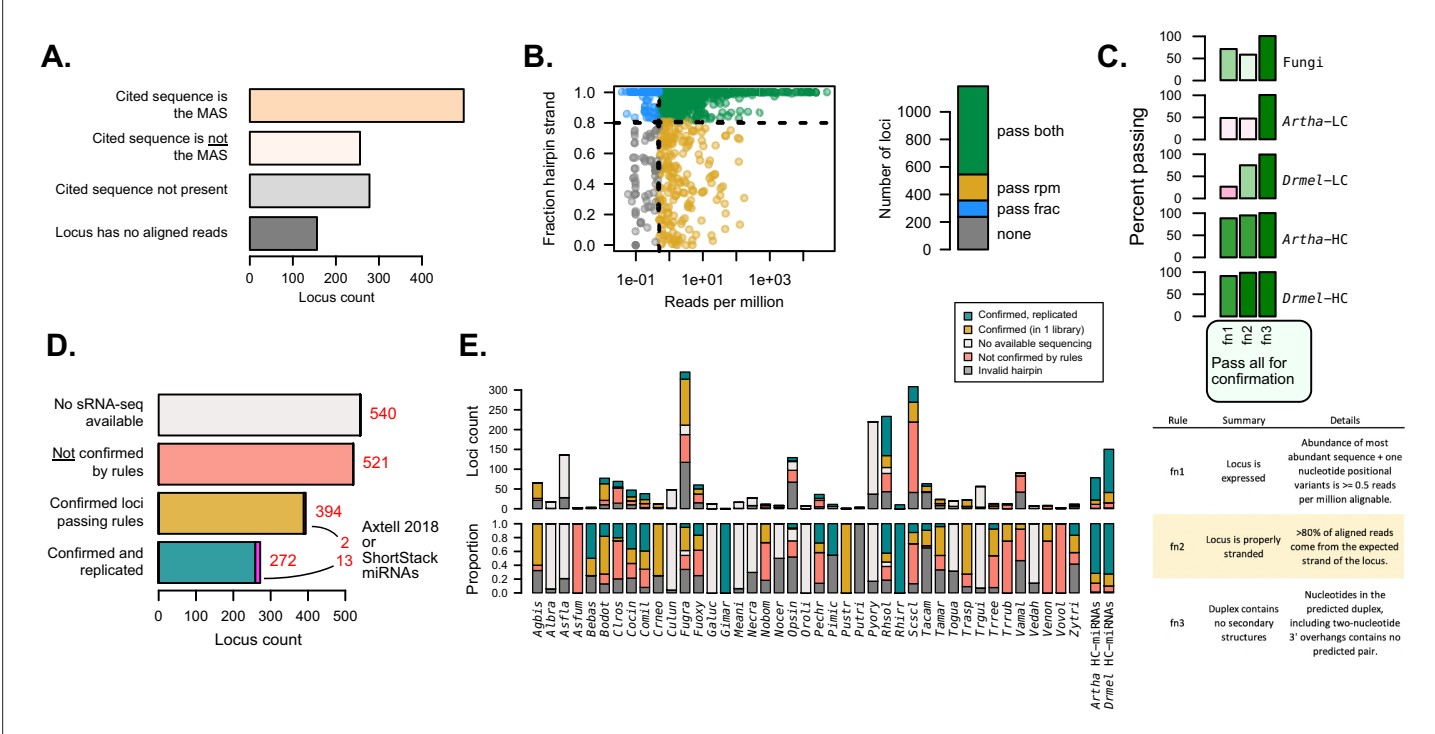

**Figure 2.** Independent evaluation of miRNAs and miRNA-like (mi/milRNA) annotations using sRNA-seq data. (**A**) Number of hairpin loci in which the reported mature mi/milRNA sequence is the most-abundant sequence (MAS) (orange), another sequence in the locus is the MAS (pink), the cited sequence is not present in the locus (light gray), or the reported locus has no aligned reads (dark gray). (**B**) Abundance (aligned reads per million) and strandedness for mi/milRNA loci. Each dot represents a locus, and the colors represent whether the locus passes the 0.5 reads per million cutoff of expression (pass rpm, yellow), has at least 80% of the aligned reads stranded (pass frac, blue), fulfills both criteria (pass both, green), or does not fulfill any of the criteria (none, gray) (**C**) Proportion of loci passing the three minimal locus profile rules (fn1, fn2, and fn3, described in table below). *Arabidopsis thaliana* (*Artha*) and *Drosophila melanogaster* (*Drmel*), low (LC) and high-confidence (HC) miRNAs from miRbase tested in a single library are included as a reference. (**D**) Number of loci without available source sRNA-seq libraries (light gray), loci failing one or more rules indicated in B and C (red), loci passing all rules (yellow), and loci passing all rules in two or more independent libraries (blue-green). For confirmed loci, we show in magenta those passing all the rules for annotating miRNAs in plants described in the ShortStack (*Axtell, 2013b*) and Axtell-2018 (*Axtell and Meyers, 2018*) rule sets. (**E**) Locus count and proportion of loci as in (**D**), separated by the parent species and including loci for which a valid hairpin could not be identified (dark gray).

The online version of this article includes the following figure supplement(s) for figure 2:

**Figure supplement 1.** Evaluation of reported miRNAs and miRNA-like (mi/milRNA) loci using published rules for plant and animal miRNAs (description of rules is provided in *Supplementary file 6*).

**Figure supplement 2.** Cumulative density function of raw library depths for source libraries reporting miRNAs and miRNA-like (mi/milRNAs).

**Figure supplement 3.** Loci count in the context of their confirmation by sRNA-seq.

This partial fulfillment of criteria is similar to the performance of miRNAs annotated as low confidence in miRbase for plants and animals (*Figure 2—figure supplement 1*). This led us to explore a less-strict set of rules to confirm that reported mi/milRNA in fungi are sRNAs derived from bona fide hairpins.

sRNAs originate from dsRNA precursors, either produced from an RNA-dependent polymerase or from a stem-loop foldback in the case of hpRNAs. These scenarios result in different alignment profiles of sRNA-seq reads, as hpRNAs will only be produced from the strand of the hairpin (*Axtell, 2013a*). It follows that the first basic criterion is that hpRNAs should largely originate from the proposed hairpin strand (>80%) (*Axtell, 2013b*; *Supplementary file 6*). A second criterion focuses on a minimum standard of expression. Here, we estimated that a minimum threshold of detection for the MAS is 0.5 reads per aligned loci (*Supplementary file 6*) as this would be at the detection limit for libraries of depths around 5 million aligned reads, a threshold passed by the raw depth of ~95% of libraries referenced in this work (*Figure 2—figure supplement 2*). Finally, we applied a third criterion for valid hairpins: the duplex for the MAS must contain no secondary structures (*Supplementary file 6*). This

needed to be re-evaluated despite the prior hairpin validation (*Figure 1E*), due to the discovery of different MASs than reported for many loci (*Figure 2A*). We found that nearly half of the loci pass these three criteria in at least one library (*Figure 2B and C*).

In order to evaluate how these three rules allowed for the discrimination between high- and low-confidence miRNA loci, we determined the performance of known miRNA reported in miRbase for *Arabidopsis thaliana* (Artha) and *Drosophila melanogaster* (Drmel) using public sRNA-seq libraries for these two organisms. We found low-confidence miRNAs performed poorly, especially in terms of strandedness and expression, while for high-confidence loci more than 80% passed all three rules (*Figure 2C*). This points to the ability of these rules to discriminate low-confidence annotations. We found that fungal loci perform considerably worse than high-confidence *Artha* and *Drmel* loci in terms of passing both expression and stranding filters (*Figure 2C*). However, it is noteworthy that these fungal loci still perform much better than low-confidence miRNA considering these metrics (*Figure 2C*), indicating that they may represent a heterogeneous mixture of sRNA, possibly including siRNA or degradation-related loci, for example.

From our initial set of 1727 loci, 394 (22.8%) passed all criteria ('confirmed' set, *Figure 2D*, *Supplementary file 4*), while 272 (15.7%) passed all criteria in two or more different libraries ('confirmed and replicated' set, *Figure 2D*, *Supplementary file 4*), for a total of 666 confirmed loci. Interestingly, 15 (0.9%) of these loci met our criteria and also the more strict set of rules used for plant miRNAs (Short-Stack and Axtell-2018) (*Axtell, 2013b*; *Axtell and Meyers, 2018*). We found important differences between the numbers of 'confirmed' and 'confirmed and replicated' loci between species, with some of them showing a lower number (e.g. *Trichophyton rubrum – Trrub*, *Verticillium dahliae – Vedal*) or a higher number (e.g. *Rhizophagus irregularis – Rhirr*, *Gigaspora margarita – Gimar*) of confirmed loci than *Artha* or *Drmel* (*Figure 2E*). Confirmation rates are highly variable in relation to source publication (*Figure 2—figure supplement 3A*), with lower confirmation rates tending to be associated with specific tools and pipelines (*Figure 2—figure supplement 3B*). While most mi/milRNA loci produce 20–24 nucleotide sRNAs as the MAS, there is no clear association of sRNA sizes with rates of confirmation (*Figure 2—figure supplement 3C*). In particular, species with high counts of identified loci such as *Fugra* and *Scscl* are prone to low confirmation rates pointing to systematic issues with the annotation pipelines, including problems with identifying valid hairpins (*Figure 2E*). While this minimal rule set is useful for filtering the lowest-confidence loci, it is likely not sufficient to form the basis of an annotation and this analysis further confirms the need for a standardized pipeline and set of criteria for miRNA annotation in fungi.

## mi/milRNA are frequently associated with protein-coding regions

The genomic context of miRNAs is an important feature to understanding their function and evolution. In animals particularly, miRNAs can associate quite closely with protein-coding genes (PCGs), originating mostly from introns, and untranslated regions (*Han et al., 2009*; *Westholm and Lai, 2011*). Conversely, plant miRNAs are mostly distant from coding regions (*Axtell et al., 2011*). In fungi, the genomic origins of mi/milRNAs have not been thoroughly studied. To gain insight into the genomic context of fungal mi/milRNAs, we first determined whether loci overlapped with any known structural or non-regulatory RNA. Structural RNAs (rRNAs, tRNAs) and non-regulatory sRNAs (snRNAs, snoRNAs) have fold patterns that might be similar to a hairpin. Considering that these are expressed sequences that likely produce short RNAs due to degradation, these may be erroneously assigned as miRNAs. Similarly, transposable elements (TEs) are known to produce siRNAs (*Nicolás and Ruiz-Vázquez, 2013*). For these reasons, mi/milRNAs that overlap with other gene products should be treated with great scrutiny. We performed a blast search of all hairpin sequences against Rfam RNA families excluding RNAi-related sRNA families (*Griffiths-Jones et al., 2003*). As shown in *Figure 3A*, most hairpins did not find hits in Rfam. The minority that did find hits can represent an outsized presence genomically. This appears to explain some of the most repetitive sequences such as bba-milR4 in *Beauveria bassiana* (Bebas), which matches the nearly universal U6 snRNA (*Figure 3—figure supplement 1*, *Figure 3A*). In this case, we find that bba-milR4 matches hundreds of loci in animals and plants (*Figure 3—figure supplement 1A*) and is found in the majority of other searched genomes (*Figure 3—figure supplement 1B*).

Hairpins with no Rfam hits were then analyzed in the context of their genomic adjacency to other genes. These were subjected to intersection analysis against the NCBI annotations of PCGs and a

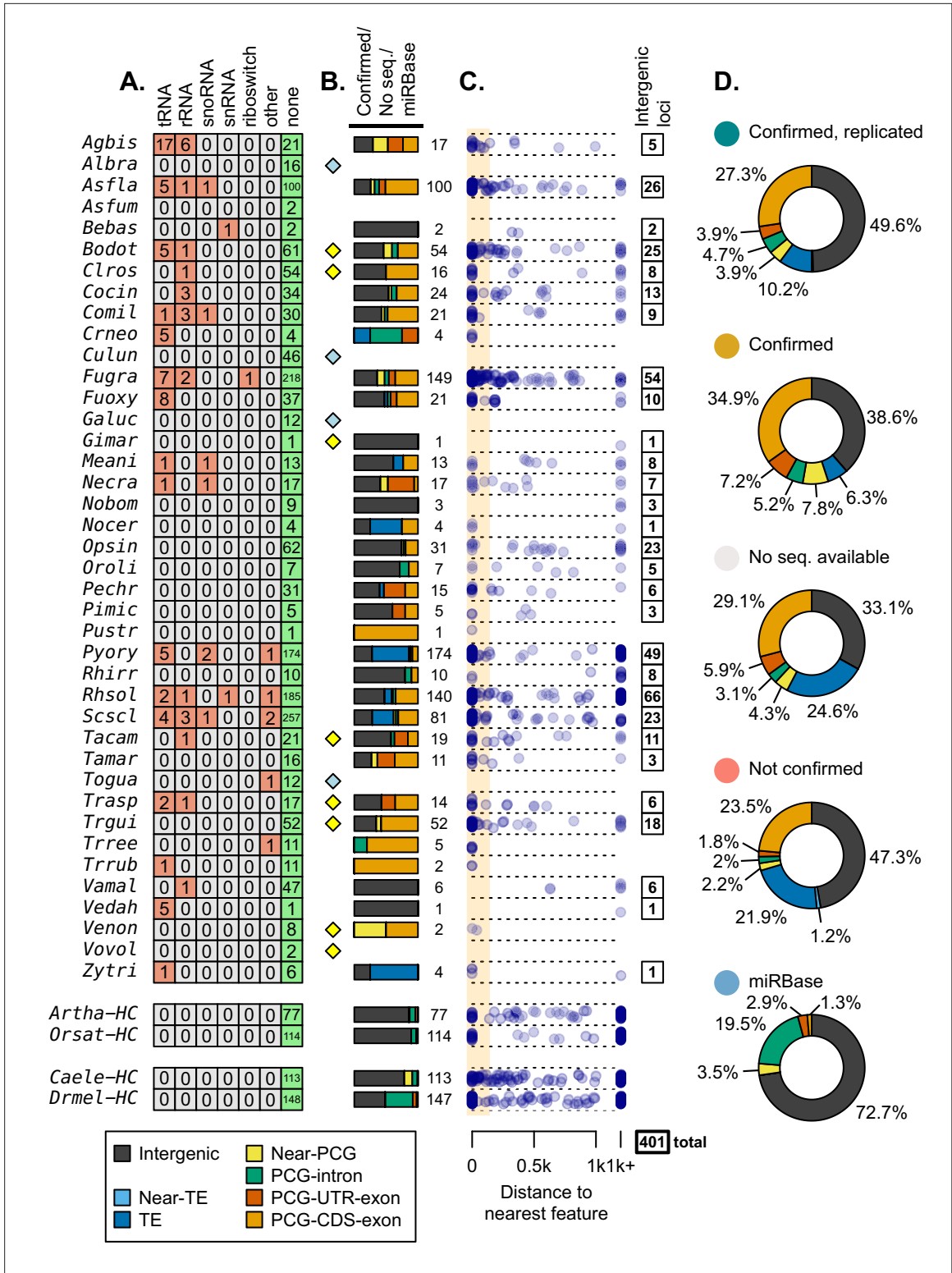

**Figure 3.** mi/milRNA loci were evaluated for homology and overlap with other genomic features. (**A**) Number of loci matching Rfam structural RNA families, identified by blastn (bitscore ≥ 50). Categories are merged from multiple Rfam entries (**Supplementary file 5**). (**B**) Proportions of loci originating from transposable elements (TEs, blue) (**Muszewska et al., 2019**), near TEs (light blue), near protein-coding genes (PCGs) (yellow), PCG introns (green), PCG untranslated regions (red), and coding sequences of PCGs (orange). Near classifications refer to distances closer than 100 nucleotides while more distant features were called as intergenic (dark gray). Proportion bars are composed of loci which are not found in Rfam, have a homologous sequence in an annotated genome, and excluding those which failed the transcriptional rules laid out in **Figure 2**. Numbers to the right of bars indicate the total

*Figure 3 continued on next page*

*Figure 3 continued*

locus count for a division. Diamonds indicate species which have no available TE annotations (yellow) or no TE/gene annotations (blue) available. (**C**) Genomic distances to the nearest feature (nucleotides) for loci in panel B. Those more distant than 1000 nucleotides are aligned to the right. Range for considering an near intersection (100 nucleotides) is shown with an orange box with those outside considered intergenic and counted to the right. (**D**) Total proportions of feature intersections for loci, divided by their transcriptional support (excluding values <1%).

The online version of this article includes the following figure supplement(s) for figure 3:

**Figure supplement 1.** Repetitiveness of unfiltered loci in related genomes.

high-quality annotation of TEs in fungi (*Muszewska et al., 2019*), where available (*Figure 3B*). Loci were considered to interact with a feature if their distance is less than 100 nucleotides, additionally labeling loci located entirely within a PCG intron. We found across species that many of the confirmed/no-sequencing-evidence loci arise from coding regions within PCGs (*Figure 3B*). This observation is concerning as this is very rarely observed in other organisms (*Liu et al., 2018*; *Olena and Patton, 2010*) and is likely a sign of incorrect annotations even in our confirmed groups, as is demonstrated in animal and plant controls (*Figure 3B*). Very rarely, mi/milRNAs from fungi were located entirely within an intron of PCGs as one might expect from a mirtron as seen in animals (*Ruby et al., 2007*; *Figure 3B*). In the case of *Drmel* high-confidence miRNAs, virtually all of those intersecting PCGs were found within introns, compared to fungi with only *Cryptococcus neoformans* (*Crneo*) showing similarly high rates of intronic mi/milRNAs (*Figure 3B*). While this is intriguing in the context of splice-machinery-derived miRNAs (*Westholm and Lai, 2011*), more evidence is needed to confirm splicing as the manner of processing of these *Crneo* mi/milRNAs. Mapping loci by their distance from a feature shows several species with near associations to loci that do not intersect (e.g. *Botryosphaeria dothidea* – *Bodot*), possibly pointing to a relationship that may not be directly transcriptionally linked (*Figure 3C*). TE intersections with mi/milRNA are a likely source of erroneous annotations, as these are known sources of sRNAs (other than miRNAs) associated with genome protection in fungi (*Nolan et al., 2005*). Comparing the feature intersections of loci in regard to their confirmation status, we find that a much lower proportion of confirmed loci intersect with TEs (*Figure 3D*), likely a sign of incorrect annotation.

We found 401 mi/milRNA loci that did not intersect other genomic features (TE, PCG) and also failed to find a homolog in the Rfam dataset (*Figure 3B*). This number was usually proportional to the number of loci reported for a given species. Considering that we cannot identify any genic features from which these loci derive, we termed these as intergenic loci which were further used for analyzing locus conservation.

## Intergenic fungal mi/milRNAs are conserved in related species

Conservation is strong evidence of the importance of a genomic element. Retention between species points toward purifying selection to maintain a sequence or structure. In plants and animals many miRNA loci are highly ancestral and conserved in sequence and function (e.g. *let-7* in animals, miR166 in plants) (*Cuperus et al., 2011*; *Pasquinelli et al., 2000*).

To define mi/milRNAs that are conserved between fungal species, we searched for orthologous mi/milRNAs, looking for evidence of a retained hairpin. A simple search for the MAS from an mi/milRNA between sequencing libraries from different species did not reveal any conserved sequences, leading us to explore whether hairpin sequences might be conserved. A genomic search for hairpin sequences is challenging, as large amounts of sequence variation can occur in many regions as long as they maintain the same structure (i.e. compensatory variation) for example, plant miR156 (*Figure 4—figure supplement 1*). To search for conserved hairpin sequences with high sensitivity, we used HMMER (*Wheeler and Eddy, 2013*). Assemblies of related fungi, plants, and animals were used as subjects, with hairpins regarded provisionally as homologs following a cutoff of >0.4 bits/base pair, attempting to normalize to allow for detection of shorter hairpins as is found in animals in addition to longer hairpins. As an input, we used only intergenic hairpins (confirmed or no sequencing evidence, *Figure 3C*) since this reduces the risk of homology due to genic elements not involved with the mi/milRNA. This has a cost of reduced sensitivity, as mi/milRNAs may originate from other genic elements. *Figure 4A* shows sub-clades of an 18S rRNA tree (*Quast et al., 2013*) for a set of related fungal genomes, indicating species that have published mi/milRNAs. Homologs are primarily found only in closely related

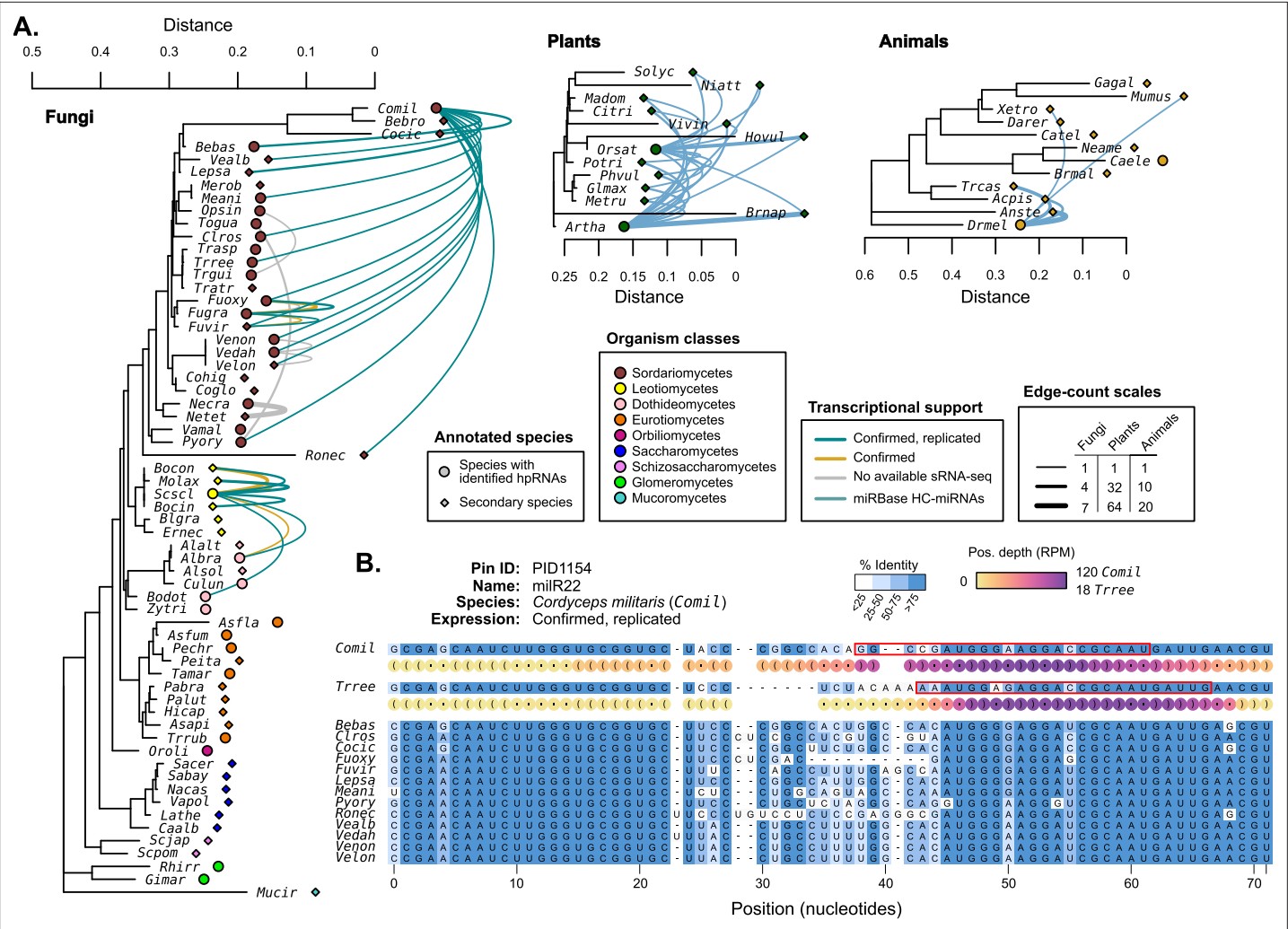

**Figure 4.** Hairpin RNAs were compared between species to identify possible homologous genes. (**A**) 18S rRNA trees (*Quast et al., 2013*) (MUSCLE alignment, RAxML maximum likelihood tree) showing selected species with published miRNAs and miRNA-like (mi/milRNA) loci (circles) and closely related species with available genomes (diamonds). Connecting edges indicate mi/milRNA loci with highly conserved hairpin sequences in other species (nhmmer, >0.4 bits/basepair). Edge colors indicate the degree of transcriptional support for the source loci in the context of *Figure 2*: green – confirmed and replicated loci, yellow – loci confirmed in a single library, gray – loci with no available sequencing libraries. HC-miRNA loci from *Arabidopsis thaliana* (*Artha*) and *Oryza sativa* (*Orsat*) are shown in blue. Edge line width is scaled to indicate the number of connections between two species, with separate scales for each clade. Edge line width indicates the number of connecting edges between two species. (**B**) Example of a conserved locus from *Cordyceps militaris* (*Comil*) with other Sordariomycetes. Bases are colored by percent identity. The hairpin structure for *Comil* and *Trichoderma reesei* (*Trree*) are shown in dot-bracket format with colored circles showing scaled depth (RPM) for all positions across available libraries. Most-abundant sequence for these species are highlighted with a red box.

The online version of this article includes the following figure supplement(s) for figure 4:

**Figure supplement 1.** Multiple sequence alignment for homologs of miR159 across plant species.

species based on the strict cutoffs mentioned, with only three examples of homology between class-level designations (*Scscl* mi/milRNAs to *Alternaria brassicicola* – *Albra*, and *Bodot*).

The most highly conserved sequence is milR22 (PID1154) from *Cordyceps militaris* (*Comil*), which is found in 15 Sordariomycetes in our assembly set (*Figure 4B*). The conservation pattern of milR22 shows a similar structure to that of known miRNAs (*Figure 4—figure supplement 1*), pointing to selection for the duplex region. Expression profiling in *Comil* shows that reads primarily come from one hairpin arm, consistent with an hpRNA locus. Profiling was also performed in the three species with putative orthologs and available sequencing data (*Bebas, Fusarium oxysporum,* and *Trichoderma reesei* - *Trree*). *Trree* was found to produce sRNAs from this putative hairpin with

a similar profile to *Comil* (***Figure 4B***), supporting that these may be orthologous genes. A closer examination of the expression profile for *Comil*-milR22 also shows very high rates of reads that only partially map to the published hairpin ('out-of-bounds', ***Source data 1***). This is found in several published mi/milRNA loci, indicating a need for a larger re-assessment of locus annotated in these publications.

While there is conservation in fungal mi/milRNAs, it does not compare with the magnitude in plants, where miRNA conservation is widespread among our examined species (***Figure 4A***). This might be an indication of the larger evolutionary distances in subject fungi compared to controls. Animal miRNAs also show signs of conservation, though distinctly lower than plants, possibly a result of their shorter hairpin lengths (***Figure 4A***). These loci with genomic conservation are strong candidates for biologically important mi/milRNAs, as selection plays a part in their retention.

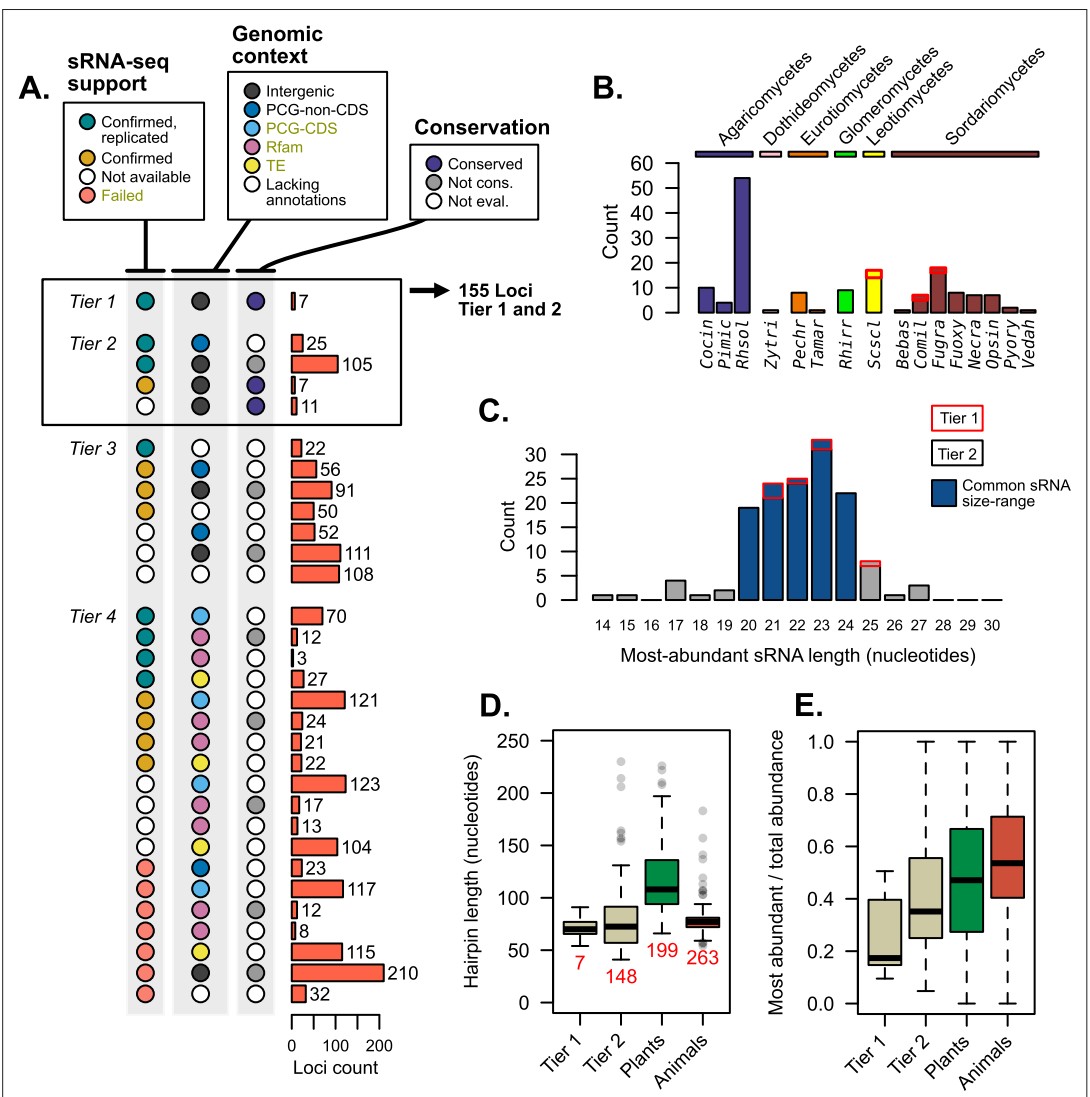

**Figure 5.** Supporting evidence was used to categorize hairpin RNAs into tier classifications. (**A**) Tier classifications of loci based on evidence quality. Colors indicate the support for a locus in the context of ***Figures 2–4***, with the bars showing the count of loci. Dark yellow labels in the legend indicate exclusion criteria (for assessment as tier 4). Tiers 1 and 2 were analyzed further, looking at (**B**) their species (showing loci count) and (**C**) the length of the most-abundant small regulatory RNA (sRNA) for a locus. Blue bars indicate size ranges common to sRNAs in other organisms. For B and C, bar outlines indicate tier of loci (red – tier 1, black – tier 2). (**D**) Hairpin lengths of tier 1 and tier 2 miRNAs and miRNA-like (mi/milRNAs) and of HC-miRNAs for plants and animals from miRbase. Length is defined as the duplex-to-duplex distance (including the duplex). (**E**) Specificity of sRNA dicing for the same groups, measured by the ratio of the most-abundant sRNA to all sRNAs matching the hairpin strand.

## Well-supported fungal mi/milRNAs have similar characteristics to plant and animal miRNAs

Each prior analyses give different facets of support to published mi/milRNA loci. Merging these diverse forms of evidence can help us categorize and rank this support, as shown in *Figure 5*. Loci were organized into four progressive tiers of support (*Figure 1—figure supplement 3*), with tier 1 being the best supported as mi/milRNAs (*Supplementary file 5*). Tier 1 hairpins are intergenic, found to be conserved between genomes, and are supported by our re-analysis of source sRNA-seq data. Tier 2 loci must have two of these three elements of support, also allowing PCG intersections not associated with a CDS in addition to intergenic loci. Tier 3 loci fail the prior two tests, but do not contain any exclusionary evidence, such as intersections with a TE or PCG-CDS, a hit with a structural RNA (Rfam), or failure to confirm with sRNA-seq evidence. These loci need more evidence to be supported as mi/milRNAs. Tier 4 loci have one of these exclusion criteria, and are likely non-mi/milRNA-derived sRNAs or represent the product of RNA degradation.

Loci falling into tiers 1 and 2 were considered to have strong support as mi/milRNA loci and were compared to miRNAs in other organisms; 155 loci met this distinction, coming from 16 different fungi from diverse clades (*Figure 5B*). The most commonly identified species was *Rhizoctonia solani* (*Rhsol*), which had the highest quantity of confirmed and replicated loci of all species (*Figure 2E*). These top-tier loci tend to fall into similar size ranges as sRNAs found in plants and animals (*Figure 5C*), though these sizes are usually more concise for miRNAs (20–22 nucleotides). Cleavage by dicer proteins is known to be the causative factor in sRNA size and only rare evidence exists for mi/milRNAs with pathways alternative to DCR (*Lee et al., 2010*). Considering this, loci falling into more atypical ranges (shorter than 19 or longer than 25 nucleotides) may be false positives for mi/milRNA as they might not be indicative of a DCR-derived locus. This is further evidence of a need of standards defining sRNA profiles of mi/milRNAs in fungi. Hairpin lengths tend to be short in fungal loci, much more similar to those of animals than plants (*Figure 5D*). Indeed, the inter-duplex length for loci is usually minimally sized, and less than 20 nucleotides in all. A very basic measure of the specificity of processing for an sRNA locus is the proportion of total locus abundance that is represented by the MAS. Here, we show that MAS of top-tier mi/milRNAs in fungi comprises a much lower proportion of locus reads (*Figure 5E*). This is possibly a sign of differences in the specificity of dicing in fungi or that these loci are not genuine miRNAs or milRNAs. Overall, these characteristics point to similar features to known miRNAs, which is further evidence of an ancient mechanism for miRNAs in eukaryotes.

Using the combined evidence presented in this work, we provide a centralized source of mi/milRNA annotations in fungi. This includes annotations outlined in GFF format (*Source data 2*) and complete tables describing extensive details of the loci with supporting evidence for all loci (*Supplementary file 5*). These form a valuable community resource for those looking to reference these loci, with all required details of their genomic coordinates, reference genome, sequence, sRNA profiles, expression, evidence of targeting when available, and confidence.

## Discussion

The growing number of publications focused on mi/milRNAs in fungi reflect an increasing awareness of the role these hpRNAs play. Despite the wealth of loci identified in these publications, it is evident that there are persistent problems with filtering out false identifications. Our work finds that only a fraction of these loci remain when performing independent validation. Particularly concerning are those loci failing minimal standards using their own sequencing data, pointing to a serious problem with the standards for locus identification based on sRNA-seq alignment profiles.

It is crucial to adopt a pipeline which is fully auditable for the publication of new sRNA loci, such as an miRNA. This means publishing the unabridged output of any tools used in annotation, including important details such as the sRNA profile of the locus itself (*Source data 1*). This is especially important if sequencing libraries for the study are not made available. Custom analyses and curation should be largely avoided, but if they are used, they must be explained thoroughly and demonstrate supporting evidence found with common tools. Many of the publications citing mi/milRNAs in fungi failed to share even the most basic features of the locus they have identified, sometimes only giving a single sRNA-sequence. For an annotation to be of use to the scientific community, full details of the locus should be given, including precursor coordinates, sequences, and predicted folding structures.

Plant and animal miRNAs also suffered from this lack of detail, which was a major motivation for establishing of the miRbase repository (*Griffiths-Jones et al., 2006*). Even with complete reporting, low-quality annotations have been pervasive in miRNA annotations (*Kozomara and Griffiths-Jones, 2014*; *Meng et al., 2012*), showing that this challenge is not unique to fungi. Indeed, evidence in this work points to many fungal mi/milRNAs that have qualities similar to low-confidence miRNAs (*Figure 2C*), further emphasizing a need for stricter standards for annotating these loci. We found that only a small number of loci passed all of the rules laid out for miRNAs by some of the more strict rule sets (*Figure 3D*; *Axtell, 2013b*; *Axtell and Meyers, 2018*). It is unclear if this is truly representative of the number of loci that might be defined by miRNA rules or if technical challenges are preventing their discovery. An approach more tuned to this clade may be required to assess the question more broadly.

Many tools exist for sRNA-annotation, with most focusing on miRNAs in animals and plants (*Axtell and Meyers, 2018*; *Friedländer et al., 2012*; *Kuang et al., 2019*). In fungi, at least one tool has been developed for milRNA identification (*Yao et al., 2020*), but uses a genome-free approach and doesn't address other categories of loci. The lack of reporting quality and differences in methods between species and studies shows a need for systematic analyses in fungal sRNA loci as has been performed for plants (*Lunardon et al., 2020*) to more clearly grasp the actual types of loci produced in this clade.

Dimensions of an sRNA locus can tell us a great deal about its synthetic pathway and likely function in an organism. Confirmed hairpins in fungi are much more similar to animals in length – they are very short with little to no terminal loop structure. Phylogenetic comparisons of dicer proteins in eukaryotes show fungal proteins differing vastly from plants and animals, where they are much more similar (*Bollmann et al., 2016*). Animals have a two-step mechanism for dicing hairpins, utilizing two different RNAse III proteins (DCR and Drosha) (*Axtell et al., 2011*). However, no evidence points to such a compartmentalization of dicing roles in fungi, further confounding the observed hairpin similarities. Lengths of the MASs from mi/milRNAs broadly fall into expected size ranges for eukaryote sRNAs (20–24 nucleotides). Specific DCRs are the causative factor for determining sRNA length (*Macrae et al., 2006*), but fungi have an average of only two DCR homologs in a species (*Choi et al., 2014*) and are thought to be partially redundant in some cases (*Lee et al., 2010*). This limited radiation of DCR proteins challenges how this range of lengths might be possible. Our results point to low specificity in fungal mi/milRNA dicing, which might explain the wide range of sizes observed, though this may also be a sign of clade-specific neofunctionalization of DCRs or other unknown protein factors.

As for the function of fungal sRNAs, there appears to be a higher average count of AGO genes across important clades (*Choi et al., 2014*), which might point to multiple roles in terms of transcriptional or post-transcriptional gene silencing. However, the details of AGO functions in terms of mi/milRNAs is still unclear and important questions still remain about how these loci may function in relation to sRNA length, their targeting dynamics, and the manner of which given sRNA affects the expression of target genes. These results are challenging to obtain and frequently require follow-up studies to confirm functionality and role of a proposed mi/milRNA (*Xu et al., 2022*). Proving targeting function in fungi is particularly difficult, as we have yet to develop a detailed view of the targeting dynamics, especially in terms of base-pairing requirements (*Broughton et al., 2016*; *Liu et al., 2014*) or distinct mechanisms of different AGOs (*Fang and Qi, 2016*). Targeting for mi/milRNAs in *Necra* appears to tolerate some degree of mismatching (*Lee et al., 2010*), though it is unclear what if any patterning requirements exist (i.e. seed pairing). *Our* analyses did identify that numerous mi/milRNAs may function more similarly to plants (*Chen et al., 2014*; *Dutta et al., 2019*; *Guo et al., 2021*; *Hunt et al., 2019*; *Jeseničnik et al., 2022*; *Shao et al., 2020*; *Silvestri et al., 2019*; *Wang et al., 2017b*), as this indicates directed cleavage of a target RNA. However, more evidence is needed to determine if mi/milRNAs in fungi function in a manner fully consistent with miRNAs in other organisms.

Few annotations curated by this work have high-quality evidence of targeting (*Supplementary file 5*), owing to the one-at-a-time nature of many targeting confirmation approaches. Three mi/milRNAs from tiers 1/2 had strong confirmed targeting (*Supplementary file 5*). One was shown in *Coprinopsis cinerea* (cci-milR-24, PID1279) regulating genes involved in developmental transition (*Lau et al., 2020*). The others were identified in two different strains of *Fusarium oxysporum* (*Fuoxy*) which infect tomato (Fol-milR1, PID2279) (*Ji et al., 2021*) and banana (milR-87, PID2109) (*Li et al., 2022*), and are likely paralogous loci producing the same MAS. Interestingly, their confirmed targets point to both *trans*-species and endogenous roles in regulation, targeting a tomato resistance gene (SlyFRG4) and

a *Fuoxy* glycosyl hydrolase (FOIG_15013) linked to pathogenicity, respectively. These show the importance of mi/milRNA in fungi both endogenously and in inter-species relationships.

While identifying proof of targeting is not a requirement of mi/milRNA annotation (*Meyers et al., 2008*), it is an essential facet for determining the biological role of any sRNA. Our analyses did identify numerous mi/milRNAs which are likely biologically relevant (tiers 1 and 2) and are strong candidates for future analyses of expression, biogenesis, and biological function/role. Those which have genomic conservation are particularly interesting, as conserved targeting relationships as have been observed in other organisms (*Johnson et al., 2019*; *Xia et al., 2013*) and would be further evidence of evolutionarily conserved roles. It is also important to note that validation of sRNA function is not necessarily a sign that it is derived from an mi/milRNA, as there are numerous RDR-derived sRNA classes that can likely function similarly. Lower tier loci with strong targeting evidence (*Supplementary file 5*), suggesting they are functional sRNAs that are not mi/milRNAs, as has been shown with other important fungal siRNAs (*Weiberg et al., 2013*).

Conservation of mi/milRNAs is a strong sign of their importance. Retention of a processed hairpin between species can be an indicator of selection for retention of function (*Cuperus et al., 2011*). However, even in systems with widespread conservation, many identified miRNAs are lineage-specific. These non-conserved miRNAs are often imprecisely processed and lowly expressed (*Qin et al., 2014*) making them frequently designated 'low confidence'. Many fungal mi/milRNAs presented here have these similar characteristics, with nearly all of top-tier loci shown as highly lineage-specific, based on our conservation analysis. Considering that the vast majority of species analyzed here are pathogens, this could be related to adaptation to hosts and/or a function as *trans*-species sRNAs (*Weiberg et al., 2013*). These sRNAs quickly differentiate in parasites, with evidence of evolution influenced by the host (*Johnson et al., 2019*). The most conserved locus from our work (milR22, *Comil*) is an insect pathogen (*Shao et al., 2019*), suggesting at a possible *trans*-species role. In all, much more evidence is needed to conclude that these genomically conserved mi/milRNAs are similarly expressed, processed, functional, and facilitate a similar role. Identifying the conservation of these aspects will be an important milestone in the exploration of mi/milRNAs in fungi.

The genomic context of fungal mi/milRNAs remains uncertain. Many mi/milRNA loci were shown to be derived from coding sequences of genes. These might warrant further investigation, though should be treated carefully, as inverted repeat structures like those in miRNA precursor stem loops might interfere with mRNA secondary structure or codons. While many loci were considered lower quality given their homology with TEs or structural RNAs, these still might be legitimate sources of mi/milRNA. Indeed, some work has concluded that miRNAs (*Yoshikawa and Fujii, 2016*) may be derived from rRNAs. However, RDR-derived sRNAs are more frequently associated with rRNAs (*Lambert et al., 2019*), including the qiRNA class in fungi (*Lee et al., 2009*). Some evidence even points to milRNAs in *Necra* which might span portions of tRNAs (*Yang et al., 2013*). Finding strong evidence supporting these loci would represent unique paths for biogenesis of mi/milRNAs and could give great insight into the genomic evolution of fungi (*Qin et al., 2015*). These annotations should be treated with caution, as all expressed loci produce short RNAs as a natural result of degradation. Accordingly, these require more attention to conclude that they are actually derived from a DCR protein and are a sRNA locus at all. Considering the enriched proportion of TEs in loci which did not pass confirmation (*Figure 3D*), loci discovered with these features are likely a symptom of false annotation. This is also true for those derived from protein-coding regions, though this phenomenon is much more documented (*Rearick et al., 2011*).

## Conclusion

A large number of sRNAs derived from hairpin sequences have been reported as mi/milRNAs throughout the literature. Here, we provide a complete and centralized annotation of these loci, giving all essential data for their exploration. This work identifies that many of the loci fail in basic validation of the locus structure and expression, highlighting a need for better standards of annotation in fungal mi/milRNA loci. Around 10% of loci were found to fall in the highest tiers of support, with these validations and annotations provided as a community resource. We found that these loci frequently arise from coding regions of the genome, including some from introns in genes. Genomically conserved mi/milRNAs also provide insight into the possible retention of these loci and their function. Overall, we see that fungal mi/milRNAs are similar in many ways to those of animals and

plants though it is not always clear to which they are more, varying in aspects of sRNA length, hairpin length, and mode of action. These results provide one of the first clade-wide explorations of this topic and allow greater perspective into the still-developing topics of mi/milRNAs in fungi.

## Materials and methods

### Finding published fungal mi/milRNAs

To assess the state of mi/milRNAs in fungi, we searched for publications related to 'miRNAs' and 'miRNA-like RNAs' in fungal species. We performed this search on the PubMed web interface using the following term: ('Fungi'[Text Word] OR 'Fungus'[Text Word] OR 'Fungal'[Text Word]) AND ('miRNA' OR 'milRNA' OR 'miRNA-like small RNA') NOT Review[Publication Type], yielding 238 results (July 29, 2022). These were subsequently filtered by hand as to whether they referenced these types of sRNAs in a fungal species. Publications were then assessed for whether they identify sequences or loci related to mi/milRNA, extracting all relevant sequence and coordinate information. Only discoveries based on small-RNA-sequencing were considered. Other information extracted was the accession numbers associated with sequencing repositories, any indications about the replication of a given mi/milRNA in multiple libraries, and the genome assembly used to identify loci. When assemblies were incompletely identified, the correct genome was inferred by manually matching coordinates (where available) to likely assemblies.

Support for mi/milRNA genes was analyzed in two contexts: genetic evidence supporting their synthesis pathway and functional evidence supporting targeting relationships. Confirmed genetic evidence constitutes studies that used knock-out, knock-downs, or over-expression to identify genetic dependencies for at least one mi/milRNA. For proving function of an mi/milRNA, studies that relied only on target prediction or indirect perturbations (i.e. a knock-out of essential sRNA genes) were considered weak evidence (*Pinzón et al., 2017*). Molecular evidence of cleavage (5'-RACE, degradome) or observing the effects of direct modulation of a specific mi/milRNA abundance (i.e. through target mimics, sRNA supplementation/over-expression, sRNA knock-outs) were considered strong evidence. Targeting evidence was identified both at the publication and at the individual mi/milRNA levels.

A complete record of loci was produced based on the available data, including the coordinates, chromosome/scaffold (with NCBI identifiers), strand, hairpin sequence, genome/assembly, and the published mature sRNA-seq. Where coordinate information was lacking but a hairpin sequence was available, coordinates were inferred using blastn (*Camacho et al., 2009*). Hairpins were folded using RNAfold (default options) (*Lorenz et al., 2011*). The duplex was estimated based on the pairing for the published mature sequence, including a conventional 2 nucleotide 5' overhang on the mature and star sequences. To be considered valid, a hairpin must (1) contain the reported MAS, (2) have no secondary structures present in the duplex region, and (3) have no more than 20 base positions in the duplex that do not have a paired sequence. A graphical explanation of publication search and mi/milRNA assessment criteria is included in *Figure 1—figure supplement 3*.

### Recovering incompletely reported loci

Loci which only reported a mature sequence or coordinates were submitted to a pipeline to recover a candidate hairpin sequence and genomic source. The pipeline first identified a likely locality for the hairpin. For those with mature coordinates, this was used as center of the locality. For those with only a single mature sequence provided, genome alignment of this sequence with bowtie (*Langmead et al., 2009*) was used to find candidate localities. For those with two mature sequences, genome alignments of these two sequences were intersected so that a locality must contain both. In the case of multiple localities found, they were tested in an arbitrary order (random). Testing involved dividing a locality into sequential candidate hairpins, 10 each of sizes 150, 300, and 600 nucleotides, making 30 candidates in all. These were folded and assessed for validity as shown prior. The recovered hairpin was chosen based on the most frequently identified duplex sequence among the hairpins. The chosen hairpin passing these requirements was reported for an mi/milRNA and all subsequent localities were ignored. Precision and sensitivity of the recovery pipeline was tested using the mature sequences only for loci with valid and fully reported hairpin sequences, including miRbase miRNA loci for *Artha*, *Orsat*, *Drmel*, and *Caele*.

## Expression profiling of hairpins

For those studies with publicly available data, libraries were downloaded and processed. *Artha* control libraries were used from a prior publication of the lead author (PRJNA543296 – stem tissue) (*Johnson et al., 2019*). *Drmel* control libraries were chosen from a recent submission from NCBI-SRA (PRJNA636660 – whole body). Adapters were identified from a set of commonly used Illumina and sRNA-seq adapters. Trimming was performed with cutadapt (-a [adapter_seq] –minimum-length 10 –maximum-length 50 –overlap 4 –max-n 0) (*Martin, 2011*). ShortStack (*Axtell, 2013b*) was used in alignment-only mode to perform a weighted bowtie alignment (*Langmead et al., 2009*) to the assembly. This approach includes alignments for most multi-mapping reads, choosing a single placement based on the local abundance of uniquely mapping reads (*Johnson et al., 2016*). Trim and alignment rates were confirmed to be mostly similar against reported rates in source publications, where available. All RPM calculations in this work were performed using all genomically aligned reads as the denominator.

Hairpins were then assessed according to rules from other publications and tools relating to miRNA identification (*Axtell and Meyers, 2018*; *Johnson et al., 2016*; *Kozomara and Griffiths-Jones, 2014*; *Kuang et al., 2019*). The rules outlined in these works are summarized in *Supplementary file 6*, including a basic hairpin rule set introduced by this work. These rules were tested on the expression profiles of small RNAs aligned to the hairpins, which do not exceed the bounds of the hairpin sequences. In some cases, the mature sequence was not the MAS in the locus.

## Homology to structural RNAs

To determine if loci are derived from a structural RNA, we used the Rfam database (*Griffiths-Jones et al., 2003*). First, we filtered a set of Rfam entries looking for families which contain a reference to at least one of the fungi with published mi/milRNA (*Supplementary file 5*). For the figures, families were generalized to categories and any annotations of small RNA loci were not considered as a structural RNA. Using this filtered list, we performed a blastn (*Camacho et al., 2009*) search for every hairpin sequence, considering hits with a bitscore ≥50.

## Genomic adjacency

To find if loci are derived or near to PCGs or TEs, we relied on annotations of these features. Assemblies and associated gene annotations for this step were all obtained from NCBI genbank accessions (with a GCA prefix). Gene annotations were filtered to only PCGs ('gene_biotype = protein coding'). TE annotations were provided from *Muszewska et al., 2019*. When the assembly associated with an mi/milRNA did not have an NCBI entry, or failed to have one or both of the annotations, we chose the most complete assembly that did. We used blastn alignment of the hairpin sequence to translate between genomes where necessary. Genomic distance was determined using bedtools closest (*Quinlan and Hall, 2010*), reporting the closest feature to the hairpin locus on either strand. This was repeated for TE and PCG annotations. In the case of a tie, a single closest feature is reported, with TEs favored in the case of a tie in distance (or intersect). Distances of 100 nucleotides or less were considered intersections in terms of calling intergenic loci. To assess intragenic context, we looked for overlaps with exon and CDS annotations, classifying a hairpin as such if over 15% of its length is derived from one of these regions. Non-exonic hairpins derived from within the gene are identified as introns, while exonic hairpins without CDS overlap are identified as coming from a UTR.

## Finding orthologous hairpin sequences

To identify putative orthologs of mi/milRNA in other fungi, we utilized genomic sequence searches. Subject genomes included all species with published mi/milRNAs shown in this study. In addition, we also included many close and distant relative species with annotated assemblies available in NCBI, especially in clades where the prior assemblies had limited resolution. Sequence searches were performed using hmmer (*Wheeler and Eddy, 2013*). Databases from assemblies were formed with the command makehmmerdb –sa_freq 2. All intergenic hairpins were then searched against the databases using the command nhmmer –tblout. Search output was filtered to include only orthologous hits (excluding hits in the same species) and redundant hits in a species, retaining only the best candidate by bitscore. To control for length of hairpins, positive hits were filtered to have a bitscore/input

sequence length >0.4. Profiles for milR22 homologs in other species were performed by aligning all libraries of said species to their hairpin profile, using bowtie (*Langmead et al., 2009*).

## Building phylogenetic trees

To compare organismal taxonomy, phylogenetic trees were constructed based on 18S ribosomal data from the SILVA database (v138.1) (*Quast et al., 2013*). Single sequences for each species were obtained from the reference NR99 aligned dataset when available and taken from the partial database when not available there. Alignments were performed to remove gaps and align the partial sequences to the NR99 alignments using MUSCLE (3.8.31) with default settings (*Edgar, 2004*). Maximum likelihood trees were produced for each organismal kingdom separately (fungi, plants, and animals) using raxml -m GTRCAT -p 9182 (*Stamatakis, 2014*).

## Tier classification criteria

Loci were classified into four confidence tiers based on metrics of sRNA-seq confirmation, genomic context, and inter-genomic conservation (*Figure 1—figure supplement 3*). Tier 1 loci are categorized to pass all criteria: they are confirmed and replicated by sequencing analysis, are derived from intergenic regions, and are conserved in two or more genomes. Tier 2 loci meet two out of three of these standards, also allowing genomic intersections with non-CDS-PCG regions (near-PCG, PCG-UTR, and PCG-intron). Tier 3 loci do not meet the higher standards of tiers 1 and 2, but also do not meet any exclusion criteria, defined as: having genomic intersection with a TE or PCG-CDS, having homology to structural RNAs (Rfam), or having failed confirmation based on sRNA-seq re-analysis. Tier 4 loci are those that failed in one or more of these exclusion criteria.

## Acknowledgements

José David Fernandez and Evelyn Sánchez (Centro de Genómica y Bioinformática, Universidad Mayor) and Dr Matthew Hasenjager (National Institute for Mathematical and Biological Synthesis, University of Tennessee, Knoxville) for help in accessing source papers and supplemental data. Dr Anna Muszewska (Institute of Biochemistry and Biophysics, Polish Academy of Sciences) for providing the complete set of TE annotations referenced in their publication (*Muszewska et al., 2019*). Dr Michael Frietag (Center for Genome Research and Biocomputing, Oregon State University) for providing the provisional genome sequences used in their publication (*Lee et al., 2010*). Dr Yulong Wang (Anhui Agricultural University) for providing details into the methodology for their publication (*Shao et al., 2019*). Funding: This work was funded by the National Agency for Research and Development of Chile (ANID) FONDECYT program (11220727 – NRJ). EAV, JMA, LFL, and NRJ were supported by ANID – Millennium Science Initiative Program – Millennium Institute for Integrative Biology (ICN17_022). LFL was supported by the Howard Hughes Medical Institute (International Research Scholar program).

---

## Additional information

### Competing interests

Luis F Larrondo: Reviewing editor, *eLife*. The other authors declare that no competing interests exist.

### Funding

| Funder | Grant reference number | Author |
|---|---|---|
| Agencia Nacional de Investigación y Desarrollo - FONDECYT | 11220727 | Nathan R Johnson |
| Agencia Nacional de Investigación y Desarrollo - Millennium Science Initiative Program | Instituto Milenio de Biologia Integrativa (iBio) - ICN17_022 | Elena A Vidal Luis F Larrondo Nathan R Johnson José M Álvarez |
| Howard Hughes Medical Institute | International Research Scholar Program | Luis F Larrondo |

| Funder | Grant reference number | Author |
|---|---|---|

The funders had no role in study design, data collection and interpretation, or the decision to submit the work for publication.

## Author contributions

Nathan R Johnson, Conceptualization, Data curation, Formal analysis, Funding acquisition, Validation, Investigation, Visualization, Methodology, Writing - original draft, Writing – review and editing; Luis F Larrondo, Supervision, Funding acquisition, Writing – review and editing; José M Álvarez, Elena A Vidal, Conceptualization, Resources, Supervision, Writing – review and editing

## Author ORCIDs

Nathan R Johnson http://orcid.org/0000-0002-5279-9964
Luis F Larrondo http://orcid.org/0000-0002-8832-7109
José M Álvarez http://orcid.org/0000-0002-5073-7751
Elena A Vidal http://orcid.org/0000-0002-8208-7327

## Decision letter and Author response

Decision letter https://doi.org/10.7554/eLife.83691.sa1
Author response https://doi.org/10.7554/eLife.83691.sa2

## Additional files

### Supplementary files

• Supplementary file 1. List of publications reporting miRNAs and miRNA-like (mi/milRNA) from fungi referred in this work. We show the publication identifier used in this work, the organism it pertains to, the publication title, journal, URL, whether the miRNA/milRNA reported were obtained by sRNA-seq analysis (Y:yes; N:no; other context), details of why a study was excluded, whether sRNA-seq data is available (Y: yes, N:no), whether mi/milRNA loci coordinates are provided (Y:yes, N:no), whether biological replication of results has been performed (Y:yes, N:no), whether genetic evidence is provided (Y:yes; N:no), type of targeting evidence provided (prediction, degradome, 5' RACE, in vitro support for targeting, and detection of target gene modulation through indirect approaches knock-outs of biosynthetic genes and direct approaches which modify the expression of the functional sRNA), and bioinformatic tool used for mi/milRNA identification. This also gives information on the native assembly used for the primary identification of the mi/milRNA loci cited in a publication, including assembly sources, IDs, names, and URLs where applicable.

• Supplementary file 2. List of species considered in this work. We show species abbreviations (abbv), full names (species), taxon identifiers (taxid), secondary names (synonyms), kingdom, phylum, class, order, family, genus, whether they are a species reported miRNAs and miRNA-like (mi/milRNAs) analyzed in this work (1: yes; 0: no), relevance to humans, hosts for pathogens/parasites, corresponding genome assemblies used for genomic context and homolog identification (ncbi_assembly), SILVA-db accessions, and whether a genus member was used as a SILVA-db substitute in the case of missing species data.

• Supplementary file 3. The sRNA-seq libraries used for re-assessment of reported miRNAs and miRNA-like (mi/milRNA) loci. We show the publication citations used in this work, SRR accession of the libraries, adapter sequence used for trimming, read length of the library, number of reads in the raw library (raw_depth), and the percentage of raw reads in the following categories: trimmed with adapters (have_adap), reads passing cutadapt filters (pass_filter), uniquely mapping reads, multi-mapping reads, percent non-mapping reads, and aligned reads resulting from unique and multi-mapper placement (aligned). Where available, this also gives the alignment rates given by source publications and a manually curated note.

• Supplementary file 4. Rules testing data for miRNAs and fungal libraries. We show identifiers for the miRNAs and miRNA-like (mi/milRNA) locus, citation, and source sRNA-seq library. Locus abundance, hairpin strand abundance, and percent stranding are shown. Other features include the abundance and sequence for the most-abundant sequence determined in this work (MAS), the inferred star for this MAS (MAS*), and the MAS cited in the source publication. Locus and cited MAS designation shown in accordance with *Figure 2A*. Rule passing is shown as a string, with the rule citation shown with a two-letter prefix (from *Supplementary file 6*), with the subsequent string pointing to rules passing (1) and failing (0) in numerical order.

• Supplementary file 5. Table defining the confirmation status for a miRNAs and miRNA-like (mi/milRNA) for all reported loci. We show identifiers for the locus, publication, native and NCBI assemblies used in its analysis, hairpin coordinates used for each assembly, and the most abundant sequence, indicating whether it has been corrected by our sequencing re-analysis. For loci with strong support for a targeting relationship, the type of evidence is shown. Locus details like the hairpin sequence, folding, minimum free energy, and boundary coordinates of the duplex within the hairpin are provided. We report identification details, including the level of coordinate and locus detail provided by the source publication and a locus's validation status. Confirmation details include the rule set confirmation status, interactions with Rfam (including Rfam descriptions), intersections with genomic features, the status as an intergenic locus, the intergenomic conservation status, and the tier classification of all reported loci. Relevant figures are indicated where available for data columns. Dashes ('-') indicate fields that are not evaluated due to an 'invalid' hairpin.

• Supplementary file 6. The miRNA annotation rules used in this work. Rule identifiers use the prefixes 'ax' (*Axtell and Meyers, 2018*), 'ku' (*Kuang et al., 2019*), 'mb' (*Kozomara and Griffiths-Jones, 2014*), and 'ss' (*Axtell, 2013b*). General rule category and more detailed rule descriptions are presented for each rule.

• MDAR checklist

• Source data 1. Text file of the sRNA abundance profiles for all miRNAs and miRNA-like (mi/milRNA) with available sRNA-seq data. Here is given the locus and library identifiers, abundance of the locus and all sRNAs aligned to the hairpin strand (l=length, a=abundance). Most-abundant sequence (MAS) and cited MASs are shown for each locus. Lower case nucleotides indicate mismatches with the reference.

• Source data 2. Annotations of miRNAs and miRNA-like (mi/milRNAs) (gffs). Includes annotations for mi/milRNAs in native genomes (where they were first annotated) and best-blast-hits (bbh) in NCBI-sourced assemblies. Annotations also contain two versions, one with all mi/milRNA loci and a second with only top tier loci (tiers 1 and 2, denoted 'best').

## Data availability

Sequencing data used in this work is available in public repositories, with publication details provided in Supplementary file 1 and all data accessions provided in Supplementary file 3. Results of abundance profiling are found in Source data 1 and summarized in Supplementary file 4.

The following previously published datasets were used:

| Author(s) | Year | Dataset title | Dataset URL | Database and Identifier |
|---|---|---|---|---|
| Johnson NR, dePamphilis CW, Axtell MJ | 2019 | Small RNA-seq from multiple Cuscuta species parasitizing Arabidopsis | https://www.ncbi.nlm.nih.gov/bioproject/PRJNA543296 | NCBI BioProject, PRJNA543296 |
| The Chinese University of Hong Kong | 2020 | Effect of formaldehyde in *Drosophila melanogaster* using transcriptomic analyses | https://www.ncbi.nlm.nih.gov/bioproject/PRJNA636660 | NCBI BioProject, PRJNA636660 |
| Zhou J, Fu Y, Xie J, Li B, Jiang D | 2012 | Identification of miRNA-like RNAs in a plant pathogen fungus Sclerotinia sclerotiorum by High-throughput sequencing | https://www.ncbi.nlm.nih.gov/bioproject/?term=PRJNA140539 | NCBI BioProject, PRJNA140539 |
| Kang K, Zhong J, Jiang L, Liu G, Gou CY | 2013 | Identification of microRNA-like RNAs in the filamentous fungus Trichoderma reesei by Solexa sequencing | https://www.ncbi.nlm.nih.gov/bioproject/?term=PRJNA201504 | NCBI BioProject, PRJNA201504 |
| Chow W-N, Wong AYP, Yeung JMY, Bao J, Lau SKP | 2013 | Penicillium marneffei Small RNA Transcriptome (17-30nt) | https://www.ncbi.nlm.nih.gov/bioproject/?term=PRJNA207279 | NCBI BioProject, PRJNA207279 |
| Chen R, Jiang N, Jiang Q, Sun X, Wang Y | 2014 | small RNA and degradome sequencing in Fusarium oxysporum | https://www.ncbi.nlm.nih.gov/bioproject/?term=PRJNA232807 | NCBI BioProject, PRJNA232807 |

*Continued on next page*

*Continued*

| Author(s) | Year | Dataset title | Dataset URL | Database and Identifier |
|---|---|---|---|---|
| Chen Y, Gao Q, Huang M, Liu Y, Liu Z | 2014 | Fusarium graminearum strain:HN9-1 Transcriptome or Gene expression | https://www.ncbi.nlm.nih.gov/bioproject/?term=PRJNA253151 | NCBI BioProject, PRJNA253151 |
| Dahlmann TA, Kück U | 2014 | Small RNA sequencing of Penicillium chrysogenum P2niaD18 | https://www.ncbi.nlm.nih.gov/bioproject/?term=PRJNA270038 | NCBI BioProject, PRJNA270038 |
| Lin Y-L, Lin S-S, Wang S-Y, Lee YR, Ma LT | 2014 | Taiwanofungus camphoratus Transcriptome or Gene expression | https://www.ncbi.nlm.nih.gov/bioproject/?term=PRJNA268267 | NCBI BioProject, PRJNA268267 |
| Yang F | 2014 | Zymoseptoria tritici small RNA transcriptome | https://www.ncbi.nlm.nih.gov/bioproject/?term=PRJNA271281 | NCBI BioProject, PRJNA271281 |
| Lin R, He L, He J, Qin P, Wang Y | 2015 | Exploring pathogenic microRNAs of rice sheath blight pathogen | https://www.ncbi.nlm.nih.gov/bioproject/?term=PRJNA282111 | NCBI BioProject, PRJNA282111 |
| Wang S, Li P, Zhang J, Qiu D, Guo L | 2015 | RNA interference is an antiviral immune response in hypovirus-infected Fusarium graminearum | https://www.ncbi.nlm.nih.gov/bioproject/?term=PRJNA304218 | NCBI BioProject, PRJNA304218 |
| Jiang X, Qiao F, Long Y, Cong H, Sun H | 2016 | microRNA-like RNAs exploring and expression profile in plant pathogenic fungus Fusarium oxysporum f. sp. niveum | https://www.ncbi.nlm.nih.gov/bioproject/?term=PRJNA329032 | NCBI BioProject, PRJNA329032 |
| Wang B, Sun Y, Song N, Zhao M, Liu R | 2016 | Puccinia striiformis f. sp. tritici strain:CYR32 Raw sequence reads | https://www.ncbi.nlm.nih.gov/bioproject/?term=PRJNA355964 | NCBI BioProject, PRJNA355964 |
| Cheng X, Cheng CK, Nong W, Cheung MK, Lau AYT | 2018 | Small RNA sequencing of Coprinopsis cinerea | https://www.ncbi.nlm.nih.gov/bioproject/PRJNA477255/ | NCBI BioProject, PRJNA477255 |
| Wang L, Xu X, Yang J, Chen L, Liu B | 2018 | Integrated microRNA and mRNA analysis in T. rubrum | https://www.ncbi.nlm.nih.gov/bioproject/?term=PRJNA483837 | NCBI BioProject, PRJNA483837 |
| Zeng W, Wang J, Wang Y, Lin J, Fu Y | 2018 | small RNA-seq at 7 day past self-fertilization in Fusarium graminearum strains | https://www.ncbi.nlm.nih.gov/bioproject/?term=PRJNA431527 | NCBI BioProject, PRJNA431527 |
| Cui C, Wang Y, Liu J, Zhao J, Sun P, Wang S | 2019 | Small RNA transcriptome sequencing of Anopheles stephensi before and post Beauveria bassiana infection | https://www.ncbi.nlm.nih.gov/bioproject/?term=PRJNA517599 | NCBI BioProject, PRJNA517599 |
| Silvestri A, Fiorilli V, Miozzi L, Accotto GP, Turina M, Lanfranco L | 2019 | SmallRNAome characterization of the arbuscular mycorrhizal fungus Rhizophagus irregularis | https://www.ncbi.nlm.nih.gov/bioproject/?term=PRJEB29180 | NCBI BioProject, PRJEB29180 |
| Hu W, Luo H, Yang Y, Wang Q, Hong N | 2018 | small RNAs sequencing of the Botryosphaeia dothidea strains | https://www.ncbi.nlm.nih.gov/bioproject/?term=PRJNA511629 | NCBI BioProject, PRJNA511629 |
| Shao Y, Tang J, Chen S, Wu Y, Wang K | 2018 | Sexual development of Cordyceps militaris | https://www.ncbi.nlm.nih.gov/bioproject/?term=PRJNA496418 | NCBI BioProject, PRJNA496418 |
| Dutta S, Jha SK, Prabhu KV, Kumar M, Mukhopadhyay K | 2014 | Triticum aestivum cultivar:HD2329 (bread wheat) | https://www.ncbi.nlm.nih.gov/bioproject/?term=PRJNA266709 | NCBI BioProject, PRJNA266709 |

*Continued*

| Author(s) | Year | Dataset title | Dataset URL | Database and Identifier |
|---|---|---|---|---|
| Silvestri A, Turina M, Fiorilli V, Miozzi L, Venice F | 2019 | Small RNAome characterization of Gigaspora margarita BEG34 germinated spores | https://www.ncbi.nlm.nih.gov/bioproject/?term=PRJEB35457 | NCBI BioProject, PRJEB35457 |
| Gong M, Wang Y, Zhang J, Zhao Y, Wan J | 2019 | Volvariella volvacea Raw sequence reads | https://www.ncbi.nlm.nih.gov/bioproject/?term=PRJNA594834 | NCBI BioProject, PRJNA594834 |
| Xu M, Guo Y, Tian R, Gao C, Guo F | 2019 | Adaptive regulation of pathogenic factors by microRNA-like RNA directed mRNA cleavage in plant pathogenic fungus Valsa mali | https://www.ncbi.nlm.nih.gov/bioproject/?term=PRJNA542139 | NCBI BioProject, PRJNA542139 |
| Xie Y, Cheung MK, Cheung PCK, Kwan HS, Lau AYT | 2019 | Small RNA sequencing of basidiospore and different germination stages in Coprinopsis cinerea #326 strain | https://www.ncbi.nlm.nih.gov/bioproject/?term=PRJNA560364 | NCBI BioProject, PRJNA560364 |
| Xia Z, Wang Z, Ding C, Liang Y, Kav NNV | 2020 | Characterization of microRNA-like RNAs associated with sclerotial development in Sclerotinia sclerotiorum | https://www.ncbi.nlm.nih.gov/bioproject/?term=PRJNA659617 | NCBI BioProject, PRJNA659617 |
| Meng H, Wang S, Yang W, Ding X, Li N | 2019 | Interactions between maize and its BLSB pathogen | https://www.ncbi.nlm.nih.gov/bioproject/?term=PRJNA596921 | NCBI BioProject, PRJNA596921 |
| Wang W, Zhang F, Cui J, Chen D, Liu Z | 2020 | sRNAs library sequencing from T. asperellum DQ-1 interacting with tomato roots | https://www.ncbi.nlm.nih.gov/bioproject/?term=PRJNA638238 | NCBI BioProject, PRJNA638238 |
| Piombo E, Vetukuri RR, Broberg A, Kalyandurg PB, Kushwaha S | 2021 | Dicer-dependant RNA silencing in Clonostachys rosea mycoparasitic action | https://www.ncbi.nlm.nih.gov/bioproject/?term=PRJEB43636 | NCBI BioProject, PRJEB43636 |
| Wong-Bajracharya J, Singan VR, Monti R, Plett KL, Ng V | 2018 | Pisolithus microcarpus SI14 smRNA sequencing Redo - SI14_1wk_ECM3 | https://gold.jgi.doe.gov/project?id=Gp0317539 | JGI GOLD, Gp0317539 |
| Wong-Bajracharya J, Singan VR, Monti R, Plett KL, Ng V | 2018 | Pisolithus microcarpus SI9 smRNA sequencing Redo - SI9_1wk_ECM1 | https://gold.jgi.doe.gov/project?id=Gp0317541 | JGI GOLD, Gp0317541 |
| Wong-Bajracharya J, Singan VR, Monti R, Plett KL, Ng V | 2017 | Pisolithus microcarpus R4 smRNA - R4_FLM3_Redo | https://gold.jgi.doe.gov/project?id=Gp0251344 | JGI GOLD, Gp0251344 |
| Wong-Bajracharya J, Singan VR, Monti R, Plett KL, Ng V | 2017 | Pisolithus microcarpus R10 smRNA - R10_FLM3_Redo | https://gold.jgi.doe.gov/project?id=Gp0251333 | JGI GOLD, Gp0251333 |
| Özkan S, Mohorianu I, Xu P, Dalmay T, Coutts RHA | 2014 | Small RNA profiles of mycovirus-free and mycovirus-infected Aspergillus fumigatus isolates, created using ScriptMiner adapters | https://www.ncbi.nlm.nih.gov/bioproject/?term=PRJNA261827 | NCBI BioProject, PRJNA261827 |

*Continued on next page*

*Continued*

| Author(s) | Year | Dataset title | Dataset URL | Database and Identifier |
|---|---|---|---|---|
| Liu M, Zhang Z, Ding C, Wang T, Kelly B, Wang P | 2020 | Transcriptomic analysis of extracellular RNA governed by the endocytic adaptor protein Cin1 of Cryptococcus deneoformans | https://www.ncbi.nlm.nih.gov/bioproject/?term=PRJNA629419 | NCBI BioProject, PRJNA629419 |
| Zhang H, Yue P, Tong X, Bai J, Yang J, Guo J | 2020 | mRNA-seq and miRNA-seq profiling analyses reveal molecular mechanisms regulating induction of fruiting bodies in Ophiocordyceps Sinensis [miRNA-seq dataset] | https://www.ncbi.nlm.nih.gov/bioproject/?term=PRJNA673414 | NCBI BioProject, PRJNA673414 |
| Dong Z, Zheng N, Hu C, Deng B, Fang W | 2021 | Constructed small RNA libraries from N. bombycis | https://www.ncbi.nlm.nih.gov/bioproject/?term=PRJNA760284 | NCBI BioProject, PRJNA760284 |
| Jeseničnik T, Štajner N, Radišek S, Mishra AK, Košmelj K | 2020 | Small RNA sequence reads of Verticillium nonalfalfae strains Recica91 and T2 | https://www.ncbi.nlm.nih.gov/bioproject/?term=PRJNA624041 | NCBI BioProject, PRJNA624041 |
| Li M, Xie L, Wang M, Lin Y, Zhong J | 2019 | small RNA sequences of Fusarium oxysporum f. sp. cubense TR4 | https://www.ncbi.nlm.nih.gov/bioproject/?term=PRJNA562097 | NCBI BioProject, PRJNA562097 |
| Marin FR, Dávalos A, Kiltschewskij D, Crespo MC, Cairns M | 2021 | Bioinformatic identification of potential MicroRNA-Like Small RNAs in the edible mushroom Agaricus bisporus and experimental approach for their validation | https://www.ncbi.nlm.nih.gov/bioproject/?term=PRJNA770841 | NCBI BioProject, PRJNA770841 |
| H-M Ji, Mao H-Y, Zhang Z-Y, Feng T, Li SJ | 2021 | Fol-milR1, a pathogenicity factor of Fusarium oxysporum, confers tomato wilt disease resistance by impairing host immune responses | https://www.ncbi.nlm.nih.gov/bioproject/PRJNA723916 | NCBI BioProject, PRJNA723916 |

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
