## [Editor Report]

This article is of interest to scientists within the field of RNA silencing and evolution. The data analysis is rigorous, and the conclusions are justified by the data. The key claims of the manuscript provide a compelling approach to identifying and annotating microRNAs in fungi.

---

## [Decision Letter]

**Decision letter after peer review:**

Thank you for submitting your article "Comprehensive re-analysis of hairpin small RNAs in fungi reveals ancestral links" for consideration by *eLife*. Your article has been reviewed by 3 peer reviewers, including Pablo A Manavella as Reviewing Editor and Reviewer #1, and the evaluation has been overseen by Detlef Weigel as the Senior Editor.

Essential revisions:

After the consultation, we believe that all the suggestions made by the reviewers are constructive and that you are likely to be able to address them. As you can see in the reviewer's comments, there was a concern regarding the functional validation of the identified miRNAs. During the consultation, we agreed that while discussing this limitation of the study would be acceptable, it would be preferable if you can provide some level of confirmation, perhaps in form of a target prediction analysis.

*Reviewer #1 (Recommendations for the authors):*

I have only a few comments, not necessarily ordered by their importance:

Line 60-62. I would change to two "main" pathways, as there are a few alternative pathways mostly involving antisense or complementary transcripts, such as the ones producing nat-siRNA in plants.

Line 71-72. miR* cleavage activity was shown by that time in plants. You should cite them: DOI: 10.1016/j.molcel.2011.04.010, DOI: 10.1104/pp.112.207068

Line 74: In plants, miRNAs function "mostly" through…

Line 81, I would rephrase and avoid citing a retracted paper.

Line 85: the abbreviation milRNA reads to me continuously as a typo. Could you replace it with "miRNA-like"? It is not much longer, and it is clearer what it means. Probably most non-specialized readers will have a similar problem.

Can you graphically display a flowthrough of the miRNA/milRNA selection criteria?

I may be missing something, sorry. In the results from figure 3, you found a considerable amount of miRNA in the coding sequences. How does the inverted repeated (IR) sequence affect the coding region? It is not common to find IR sequences in coding regions.

I'm curious about figure 5 C. Contrary to your statement, this is not a typical profile for miRNAs (especially in plants). Most miRNAs in plants are 21 nt with a few 20 and 22, but almost no 23-24, which DCL3 produces from TE. There are some exceptions, for example, in dcl1 mutants or in flowers where DCL3 may overcome DCL1 function and produce 23-24 "miRNA", but in these cases, these sRNA do not work as miRNA but as siRNA triggering TGS instead of PTGS. Is there any evidence that 24 nt siRNA act in PTGS in fungi? You mentioned that fungi have 2 DCR proteins. What about AGO? as PTGS in plants is mainly triggered by 21 nt loaded into proteins from the AGO1 clade and TGS by 24 loaded into AGO4 clade.

*Reviewer #2 (Recommendations for the authors):*

1. Figures contain too much detailed information from the statistical analysis, which cannot be easily interpreted independently of the main text. It is suggested to show the key points of data for better understanding.

2. It is suggested to draw the biogenesis process of fungal mi/milRNAs in the figures, including their genomic loci (exon and intron of protein-coding genes, TE, intergenic regions) and hairpins to better show their features.

3. How fungal mi/milRNA loci are classified into four tiers? Please state the classification criteria.

*Reviewer #3 (Recommendations for the authors):*

The absence of follow-up experiments somewhat limits the impact of this paper. Overall this paper will be of interest to readers trying to understand the characteristics and functions of miRNA and miRNA-like hairpins in fungi. Though the theme is interesting, I felt some clarification is needed. I listed some critical points to be reconsidered. I hope it helps the authors improve the manuscript.

– l:364 You mentioned that you use HMMER to search for conserved hairpin sequences. You use a cutoff of > 0.4 bits/base pair, attempting to normalize for shorter hairpins, similar to animals.

But, in Figure 5D in tier2 despite, on average the length of the hairpin being even below that of animals, you have several outliers with larger hairpins (similar to plants). So, why try to center only in shorter hairpins? I don't see the point here.

– l:400 "While there is conservation in fungal mi/milRNAs, it does not compare with the magnitude in plants, where miRNA conservation is widespread"

This is partially true, at least by the phylogenetic trees you are showing. In plants, you are comparing only angiospers (mostly eudicots plants). But, if you compare less closely related plants, for example, Artha with some moss (bryophytes), you won't find widespread miRNA conservation. The evolutionary distance in fungi you are comparing seems higher than in plants. Thus, to support your claim, it would be interesting to compare two similar evolutionary distant groups in fungi and plants. Or at least mention it in the discussion.

– I already mentioned that the absence of follow-up experiments somewhat limits the impact of this paper. I completely understand that this work is mostly computational. But, exploring the potential targets of miRNAs is one of the main aims when studying miRNAs in different species. I suggest looking for potential targets for the strong mi/milRNAs candidates detected in your work.

It would be nice to discuss the most important rules to predict miRNA targets in fungi. (More similar to plants or animals?)

---

## [Author Response]

Essential revisions:After the consultation, we believe that all the suggestions made by the reviewers are constructive and that you are likely to be able to address them. As you can see in the reviewer's comments, there was a concern regarding the functional validation of the identified miRNAs. During the consultation, we agreed that while discussing this limitation of the study would be acceptable, it would be preferable if you can provide some level of confirmation, perhaps in form of a target prediction analysis.

We thank the editor for the positive evaluation of our study and the opportunity to improve and resubmit our manuscript. We agree with all of the suggestions and have made an effort to remedy them by text or re-analysis where necessary or possible. Direct responses to all comments/suggestions are shown below.

As for the essential revision of exploring targeting, we decided to use the confirmed targets proven in source publications and secondary publications referencing the mi/milRNA loci evaluated in our study. Though these represent only a small number of loci, we believe that their inclusion contributes importantly to the record of annotation presented in this work. All loci with strong validation (molecular evidence, genetic evidence) of function are now identified in our annotation table, pointing to the type of evidence used in the confirmation. This information is now available in Supplementary File 5.

We have additionally added a passage to the discussion directly addressing this topic. Here we explain that function is important for understanding what a mi/milRNA does biologically, but is not necessary for its annotation (Meyers et al., 2008), pointing to the pitfalls/shortcomings of prediction-only approaches. We also highlight what supported top-tier loci do, both in terms of their targets and roles.

We decided to not perform our own analysis of target prediction for two primary reasons:

–Our work excludes prediction-only evidence of targeting in our assessment of functional support, requiring stronger validation support. These analyses are fraught with false positives (Pinzón et al., 2017), which is further exacerbated by our lack of knowledge on complementarity requirements in fungi.

– Many fungi producing top-tier mi/milRNAs have important inter-species relationships (pathogens/parasites/mutualists), sometimes as a generalist (may be compatible with many hosts). Considering trans-species sRNAs, the host transcriptome adds a level of complexity that we believe takes this analysis out of the scope of the study.

Reviewer #1 (Recommendations for the authors):I have only a few comments, not necessarily ordered by their importance:Line 60-62. I would change to two "main" pathways, as there are a few alternative pathways mostly involving antisense or complementary transcripts, such as the ones producing nat-siRNA in plants.

Agreed, text modified to include this distinction.

Line 71-72. miR* cleavage activity was shown by that time in plants. You should cite them: DOI: 10.1016/j.molcel.2011.04.010, DOI: 10.1104/pp.112.207068

Agreed, citation added to this section.

Line 74: In plants, miRNAs function "mostly" through…

Agreed, text modified to include this distinction.

Line 81, I would rephrase and avoid citing a retracted paper.

We struggled with this. Considering that this paper is the only evidence of this phenomenon we are aware of, and that it has been subsequently retracted, we have taken the reviewer’s suggestion and removed the corresponding passage. The text now reads:

“Long hairpin RNAs have long been a biotechnological tool for induction of RNAi (Fusaro et al., 2006), however, exploration is needed to identify clearly if this occurs in organisms naturally.”

Line 85: the abbreviation milRNA reads to me continuously as a typo. Could you replace it with "miRNA-like"? It is not much longer, and it is clearer what it means. Probably most non-specialized readers will have a similar problem.

We acknowledge the reviewer’s comment, however we have chosen to maintain our naming/abbreviation schema for the work. The abbreviation “milRNA” is commonly used in fungal literature. Additionally, we believe it’s frequent use in this manuscript calls for abbreviation and leads to less confusion.

Can you graphically display a flowthrough of the miRNA/milRNA selection criteria?

We have taken this suggestion to add a flowchart as Figure 1—figure supplement 3, which explains the publication, mi/milRNA, sRNA-sequencing confirmation, and tier selection criteria.

I may be missing something, sorry. In the results from figure 3, you found a considerable amount of miRNA in the coding sequences. How does the inverted repeated (IR) sequence affect the coding region? It is not common to find IR sequences in coding regions.

This is an important question. To answer it, we re-evaluated our classification of genomic intersections to include more specific distinctions in classes (near-TE, TE, near-PCG, PCG-CDS, PCG-UTR, PCG-intron, intergenic). This re-evaluation (shown in modified figure 3) shows that much fewer miRbase annotated miRNAs are coming from coding regions: a more believable control.

However, this still shows that CDS mi/milRNA annotations are common in fungi. We believe this may be a symptom of incorrect annotations, and fits well with our strategy of using multiple analyses to classify confidence tiers. We have further corrected figure 5 (our tier classifications) to include a distinction between CDS and non-CDS PCG overlaps, where PCG-CDS overlaps are categorized in tier 4.

We have additionally added this phrase to clarify the result in this section:

“We found across species that many of the confirmed/no-sequencing-evidence loci arise from coding-regions within PCGs (Figure 3B). This observation is concerning as this is very rarely observed in other organisms (Liu et al., 2018; Olena and Patton, 2010) and is likely a sign of incorrect annotations even in our confirmed groups, as is demonstrated in animal and plant controls (Figure 3B).”

Furthermore, we added a statement to the discussion to clarify our thoughts on this topic:

“Many mi/milRNA loci were shown to be derived from coding-sequences of genes. These might warrant further investigation, though should be treated carefully, as inverted repeat structures like those in miRNA precursor stem loops might interfere with mRNA-secondary structure or codons.”

I'm curious about figure 5 C. Contrary to your statement, this is not a typical profile for miRNAs (especially in plants). Most miRNAs in plants are 21 nt with a few 20 and 22, but almost no 23-24, which DCL3 produces from TE. There are some exceptions, for example, in dcl1 mutants or in flowers where DCL3 may overcome DCL1 function and produce 23-24 "miRNA", but in these cases, these sRNA do not work as miRNA but as siRNA triggering TGS instead of PTGS. Is there any evidence that 24 nt siRNA act in PTGS in fungi? You mentioned that fungi have 2 DCR proteins. What about AGO? as PTGS in plants is mainly triggered by 21 nt loaded into proteins from the AGO1 clade and TGS by 24 loaded into AGO4 clade.

This is an excellent question. The text and the figure have now been changed to clarify that these are sRNA size ranges in plants/animals, not common miRNA sizes.

“These top-tier loci tend to fall into similar size ranges as sRNAs found in plants and animals (Figure 5C), though these sizes are usually more concise for miRNAs (20-22 nucleotides).”

As for the questions in terms of AGO and sRNA size-dependent functionality, little is known in fungi. To address this, we modified our discussion to read:

“As for the function of fungal sRNAs, there appears to be a higher average count of AGO genes across important clades (Choi et al., 2014), which might point to multiple roles in terms of transcriptional or post-transcriptional gene silencing. However, the details of AGO functions in terms of mi/milRNAs is still unclear and important questions still remain about how these loci may function in relation to sRNA length, their targeting dynamics, and the manner of which given sRNA affects the expression of target genes.”

Reviewer #2 (Recommendations for the authors):1. Figures contain too much detailed information from the statistical analysis, which cannot be easily interpreted independently of the main text. It is suggested to show the key points of data for better understanding.

We took this point in regard to figure 3, which had many numbers and clauses represented in it. Considering these data are shared in Supplementary File 5, we reduced the complexity of this figure focusing it on only the loci which did not fail the sequencing re-analysis, and we now believe it is much clearer.

2. It is suggested to draw the biogenesis process of fungal mi/milRNAs in the figures, including their genomic loci (exon and intron of protein-coding genes, TE, intergenic regions) and hairpins to better show their features.

Figure 1A now shows a simple cartoon giving context for hairpin-derived sRNAs, introducing what is known about their source regions.

3. How fungal mi/milRNA loci are classified into four tiers? Please state the classification criteria.

These criteria are now clearly stated in the figure 1-supplement 3 flowchart, as well as rephrased for clarity of the description in results.

We have also added a section to the results to clarify these:

“Tier classification criteria

Loci were classified into four confidence tiers based on metrics of sRNA-seq confirmation, genomic context, and inter-genomic conservation (Figure 1—figure supplement 3). Tier 1 loci are categorized to pass all criteria: they are confirmed and replicated by sequencing analysis, are derived from intergenic regions, and are conserved in 2 or more genomes. Tier 2 loci meet two out of three of these standards, also allowing genomic intersections with non-CDS-PCG regions (near-PCG, PCG-UTR, and PCG-intron). Tier 3 loci do not meet the higher standards of tiers 1 and 2, but also do not meet any exclusion criteria, defined as: having genomic intersection with a TE or PCG-CDS, having homology to structural RNAs (Rfam), or having failed confirmation based on sRNA-seq re-analysis. Tier 4 loci are those that failed in one or more of these exclusion criteria.”

Reviewer #3 (Recommendations for the authors):The absence of follow-up experiments somewhat limits the impact of this paper. Overall this paper will be of interest to readers trying to understand the characteristics and functions of miRNA and miRNA-like hairpins in fungi. Though the theme is interesting, I felt some clarification is needed. I listed some critical points to be reconsidered. I hope it helps the authors improve the manuscript.– l:364 You mentioned that you use HMMER to search for conserved hairpin sequences. You use a cutoff of > 0.4 bits/base pair, attempting to normalize for shorter hairpins, similar to animals.But, in Figure 5D in tier2 despite, on average the length of the hairpin being even below that of animals, you have several outliers with larger hairpins (similar to plants). So, why try to center only in shorter hairpins? I don't see the point here.

This normalization was used to allow us to detect smaller hairpins like those which are found in animals, but it should not exclude longer hairpins. We have reworded the text to be clearer that it does not “center” on short hairpins:

“Assemblies of related fungi, plants, and animals were used as subjects, with hairpins regarded provisionally as homologs following a cutoff of > 0.4 bits / base pair, attempting to normalize to allow for detection of shorter hairpins as is found in animals in addition to longer hairpins.”

– l:400 "While there is conservation in fungal mi/milRNAs, it does not compare with the magnitude in plants, where miRNA conservation is widespread"This is partially true, at least by the phylogenetic trees you are showing. In plants, you are comparing only angiospers (mostly eudicots plants). But, if you compare less closely related plants, for example, Artha with some moss (bryophytes), you won't find widespread miRNA conservation. The evolutionary distance in fungi you are comparing seems higher than in plants. Thus, to support your claim, it would be interesting to compare two similar evolutionary distant groups in fungi and plants. Or at least mention it in the discussion.

We have reworded this statement to be clearer that our methodology focuses only on our examined species. We also mention the limitation of distance in our controls.

“While there is conservation in fungal mi/milRNAs, it does not compare with the magnitude in plants, where miRNA conservation is widespread among our examined species (Figure 4A). This might be an indication of the larger evolutionary distances in subject fungi compared to controls.”

– I already mentioned that the absence of follow-up experiments somewhat limits the impact of this paper. I completely understand that this work is mostly computational. But, exploring the potential targets of miRNAs is one of the main aims when studying miRNAs in different species. I suggest looking for potential targets for the strong mi/milRNAs candidates detected in your work.

This paper focuses on mi/milRNAs, which are defined by their biogenesis. While proof of function is not considered a requirement for their annotation (Axtell and Meyers, 2018; Meyers et al., 2008). We agree that assessing targets is highly desirable for identifying the biological role of sRNAs, an interesting topic that however falls beyond the scope of this paper.

Considering we require high-quality experimental confirmation to consider targeting, we have decided not to explore our own target prediction as a way to address this comment. Prediction suffers from prohibitive false positive rates (Pinzón et al., 2017), which can only be exacerbated by the lack of knowledge we have on miRNA targeting dynamics in fungi.

Instead, we focused on cited evidence of targeting – in keeping with the re-analysis and curation presented here. To try to answer this suggestion, we have aggregated all of the confirmed targeting interactions, including the manner of evidence for all reported mi/milRNAs (where available). These data are now available in Supplementary File 5 (column “functional_support”). We believe this adds great value to any researcher looking to use our centralized annotations.

We also addressed this in the text in several ways, including a re-ordering of the final paragraphs.

1) Discussing our lack of knowledge of targeting dynamics:

“Proving targeting function in fungi is particularly difficult, as we have yet to develop a detailed view of the targeting dynamics, especially in terms of base-pairing requirements (Broughton et al., 2016; Liu et al., 2014) or distinct mechanisms of different AGOs (Fang and Qi, 2016).”

2) Discussing top-tier loci that have been confirmed (3), as well as their targets and roles:

“Few annotations curated by this work have high-quality evidence of targeting (Supplementary File 5), owing to the one-at-a-time nature of many targeting confirmation approaches. Three mi/milRNAs from Tiers 1/2 had strong confirmed targeting (Supplementary File 5). One was shown in Coprinopsis cinerea (cci-milR-24, PID1279) regulating genes involved in developmental transition (Lau et al., 2020). The others were identified in two different strains of Fusarium oxysporum (Fuoxy) which infect tomato (Fol-milR1, PID2279) (Ji et al., 2021) and banana (milR-87, PID2109) (Li et al., 2022), and are likely paralogous loci producing the same MAS. Interestingly, their confirmed targets point to both trans-species and endogenous roles in regulation, targeting a tomato resistance gene (SlyFRG4) and a Fuoxy glycosyl hydrolase (FOIG_15013) linked to pathogenicity, respectively. These show the importance of mi/milRNA in fungi both endogenously and in inter-species relationships.”

3) Discussing the importance of target confirmation while highlighting it is not essential for annotation:

“While identifying proof of targeting is not a requirement of mi/milRNA annotation (Meyers et al., 2008), it is an essential facet for determining the biological role of any sRNA. Our analyses did identify numerous mi/milRNAs which are likely biologically relevant (tiers 1 and 2) and are strong candidates for future analyses of expression, biogenesis, and biological function/role. Those which have genomic conservation are particularly interesting, as conserved targeting relationships have been observed in other organisms (Johnson et al., 2019; Xia et al., 2013) and would be further evidence of evolutionarily conserved roles. It is also important to note that validation of sRNA function is not necessarily a sign that it is derived from a mi/milRNA, as there are numerous RDR-derived sRNA classes that can likely function similarly. Lower tier loci with strong targeting evidence (Supplementary File 5), suggesting they are functional sRNAs that are not mi/milRNAs, as has been shown with other important fungal siRNAs (Weiberg et al., 2013).”

It would be nice to discuss the most important rules to predict miRNA targets in fungi. (More similar to plants or animals?)

We now highlight what is known on this topic with the following statement:

“Targeting for mi/milRNAs in Necra appears to tolerate some degree of mismatching (Lee et al., 2010), though it is unclear what if any patterning-requirements exist (i.e., seed-pairing).”

References

Axtell MJ, Meyers BC. 2018. Revisiting Criteria for Plant MicroRNA Annotation in the Era of Big Data. Plant Cell 30:272–284.

Broughton JP, Lovci MT, Huang JL, Yeo GW, Pasquinelli AE. 2016. Pairing beyond the Seed Supports MicroRNA Targeting Specificity. Mol Cell 64:320–333.

Fang X, Qi Y. 2016. RNAi in Plants: An Argonaute-Centered View. Plant Cell 28:272–285.

Johnson NR, dePamphilis CW, Axtell MJ. 2019. Compensatory sequence variation between trans-species small RNAs and their target sites. eLife 8. doi:10.7554/eLife.49750

Liu B, Shyr Y, Cai J, Liu Q. 2018. Interplay between miRNAs and host genes and their role in cancer. Brief Funct Genomics 18:255–266.

Liu Q, Wang F, Axtell MJ. 2014. Analysis of complementarity requirements for plant microRNA targeting using a Nicotiana benthamiana quantitative transient assay. Plant Cell 26:741–753.

Meyers BC, Axtell MJ, Bartel B, Bartel DP, Baulcombe D, Bowman JL, Cao X, Carrington JC, Chen X, Green PJ, Griffiths-Jones S, Jacobsen SE, Mallory AC, Martienssen RA, Poethig RS, Qi Y, Vaucheret H, Voinnet O, Watanabe Y, Weigel D, Zhu J-K. 2008. Criteria for annotation of plant MicroRNAs. Plant Cell 20:3186–3190.

Olena AF, Patton JG. 2010. Genomic organization of microRNAs. J Cell Physiol 222:540–545.

Pinzón N, Li B, Martinez L, Sergeeva A, Presumey J, Apparailly F, Seitz H. 2017. microRNA target prediction programs predict many false positives. Genome Res 27:234–245.

Weiberg A, Wang M, Lin F-M, Zhao H, Zhang Z, Kaloshian I, Huang H-D, Jin H. 2013. Fungal small RNAs suppress plant immunity by hijacking host RNA interference pathways. Science 342:118–123.

Xia R, Meyers BC, Liu Zhongchi, Beers EP, Ye S, Liu Zongrang. 2013. MicroRNA superfamilies descended from miR390 and their roles in secondary small interfering RNA Biogenesis in Eudicots. Plant Cell 25:1555–1572.